# Using a synthetic machinery to improve carbon yield with acetylphosphate as the core

Likun Guo[1,3], Min Liu[1,3], Yujia Bi[1], Qingsheng Qi [1], Mo Xian[2] & Guang Zhao [1,2] ✉

In microbial cell factory, $CO_2$ release during acetyl-CoA production from pyruvate significantly decreases the carbon atom economy. Here, we construct and optimize a synthetic carbon conserving pathway named as Sedoheptulose-1,7-bisphosphatase Cycle with Trifunctional PhosphoKetolase (SCTPK) in *Escherichia coli*. This cycle relies on a generalist phosphoketolase Xfspk and converts glucose into the stoichiometric amounts of acetylphosphate (AcP). Furthermore, genetic circuits responding to AcP positively or negatively are created. Together with SCTPK, they constitute a gene-metabolic oscillator that regulates Xfspk and enzymes converting AcP into valuable chemicals in response to intracellular AcP level autonomously, allocating metabolic flux rationally and improving the carbon atom economy of bioconversion process. Using this synthetic machinery, mevalonate is produced with a yield higher than its native theoretical yield, and the highest titer and yield of 3-hydroxypropionate via malonyl-CoA pathway are achieved. This study provides a strategy for improving the carbon yield of microbial cell factories.

Recently, bioproduction of desired chemicals and materials is getting momentum due to their environmental friendliness and practical feasibility. In microbial cell factories design and construction processes, a key issue is to improve the fluxes towards the product of interest and to maximize the carbon atom economy[1]. This is quite important especially under the global achievement of carbon peak and carbon neutrality. The core of any metabolic network in microorganism is a high-flux backbone, generally referred to as the central metabolism[2], responsible for the transformation of primary input substrates into energy and a series of building blocks for the production of cellular polymers and considered as the invariable operating system of the cell[3]. Therefore, if we want to build a synthetic machinery that convert a living organism into a truly productive biofactory, apart from optimizing the biosynthetic pathway as standalone unit, a successful bioengineering approach must also bend the endogenous metabolic network of the host, especially its central metabolism, to

support the bioproduction process[4]. In microorganism, acetyl coenzyme A (acetyl-CoA) is not only a fundamental metabolite in central metabolic pathways, but also a precursor for numerous industrially relevant products[5–7]. So, rewriting the microorganism's central metabolism to supply plenty acetyl-CoA with high carbon atom economy will benefit the bioproduction of a wide variety of chemicals and materials.

To improve the acetyl-CoA yield, a typical strategy is the elimination of carbon flux to competing pathways and the overexpression of important enzyme to ensure sufficient production of acetyl-CoA[8,9]. This approach is more straightforward and simpler in design, but does not actually increase the theoretical carbon yield of the target chemical. Natural microorganisms usually convert glucose into acetyl-CoA through the glycolysis pathway together with decarboxylation of pyruvate by the pyruvate dehydrogenase, in which 2 mol of acetyl-CoA and 2 mol of $CO_2$ are generated from each mol of glucose[10]. This

[1]State Key Laboratory of Microbial Technology, Shandong University, Qingdao 266237, China. [2]CAS Key Lab of Biobased Materials, Qingdao Institute of Bioenergy and Bioprocess Technology, Chinese Academy of Sciences, Qingdao 266101, China. [3]These authors contributed equally: Likun Guo, Min Liu. ✉e-mail: zhaoguang@sdu.edu.cn

release of $CO_2$ causes a significant decrease of the atomic economy of targeted chemical biosynthetic pathway, representing the major carbon loss in microbial carbon metabolism and biorefining[11]. To resolve this problem, a synthetic non-oxidative glycolysis (NOG) route was developed (Supplementary Fig. 1), converting glucose into the stoichiometric amounts of acetylphosphate (AcP), which is further catalyzed by phosphate acetyltransferase to acetyl-CoA[12]. This NOG configuration relies on carbon rearrangement and phosphoketolase which catalyzes the irreversible conversion of fructose-6-phosphate (F6P) or xylulose-5-phosphate (X5P) into AcP and erythrose-4-phosphate (E4P) or glyceraldehyde-3-phosphate (GAP), respectively.

Recently, based on mathematical modeling, Andersen et al. proposed a new variant of the NOG pathway with introduction of a novel phosphoketolase activity towards sedoheptulose-7-phosphate (S7P)[13]. Subsequently, Hellgren et al. constructed the proposed NOG variant in *Saccharomyces cerevisiae*[14]. The phosphoketolase from *Bifidobacterium longum*, which was shown to act upon S7P, F6P and X5P (Xfspk), together with an endogenous sedoheptulose-1,7-bisphosphatase (SBPase), were overexpressed to form a sedoheptulose-1,7-bisphosphate (SBP) -dependent cycle pathway (Fig. 1 and Supplementary Fig. 1). Similar with NOG pathway, this route converts each mol glucose into 3 mol AcP with consumption of 1 mol ATP, and there is no net production or consumption of reducing power (Supplementary

Data 1). On the other hand, the SBP-dependent cycle is the shortest carbon conservation pathway, requiring only 6 enzymes while 8 enzymes in NOG. It no longer requires transaldolase and transketolase for carbon rearrangement as NOG pathway, reducing the number of involved enzymes and the complexity of this pathway, representing the carbon atom economy as well as the protein economy. However, that study was just a proof-of-concept and engineered yeast strains carrying the SBP-dependent cycle still presented low yield of acetyl-CoA derived chemicals from glucose[14]. Besides *S. cerevisiae*, *Escherichia coli* is also a host microorganism of choice for bioproduction. In this study, this SBP-dependent cycle is named after its characteristic intermediate and enzyme as SBP Cycle with Trifunctional Phospho-Ketolase (SCTPK), and will be evaluated and optimized in *E. coli*.

The output of the SCTPK is AcP, which is not only a key regulatory node for central metabolism, but also a global signal affecting diverse cellular processes via protein post-translational modification[15]. AcP can donate its phosphoryl group to and activate response regulators of two-component signal transduction systems[16,17], and also can acetylate lysine residue to modulate the structure, enzymatic activity, and stability of corresponding protein[18,19]. To avoid possible unfavorable effects caused by AcP accumulation, it is essential to develop an AcP-responsive genetic circuit to adjust the expression level of genes related in AcP production and utilization sophisticatedly in an

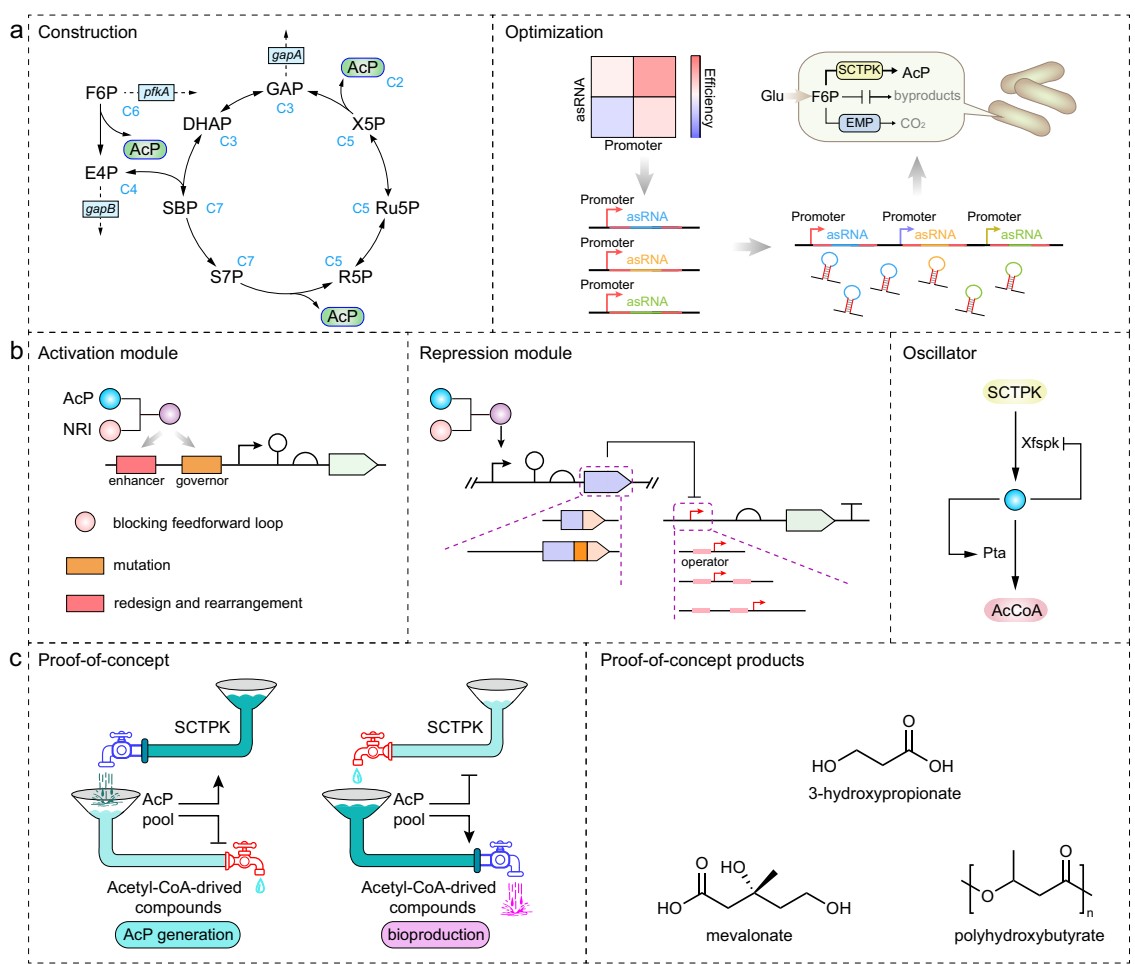

**Fig. 1 | The framework of the synthetic machinery to improve carbon yield with acetylphosphate as the core. a** Decreasing $CO_2$ emissions through the construction and optimization of a carbon conservation pathway SCTPK. G6P glucose-6-phosphate, F6P fructose-6-phosphate, AcP acetylphosphate, E4P erythrose-4-phosphate, SBP sedoheptulose-1,7-bisphosphate, GAP glyceraldehyde-3-phosphate, PfkA 6-phosphofructokinase I, GapB erythrose-4-phosphate dehydrogenase, GapA glyceraldehyde-3-phosphate dehydrogenase. **b** Development of AcP-centered gene-metabolic oscillator, that consists of an activation module and a repression module to dynamically reallocate flux. **c** Applications of the synthetic machinery in biosynthesis of chemicals and materials. Products for proof of concept include 3-hydroxypropionate, mevalonate and polyhydroxybutyrate.

engineered strain carrying SCTPK. The Ntr system, consisting of the sensor NRII (*glnL* product) and the response regulator NRI (*glnG* product), controls a transcriptional response to nitrogen starvation[20]. When NRII is absent, NRI itself can sense the AcP level and activate the *glnAp2* promoter accordingly[21]. As the Ntr system plays a negligible role under most bioreactor conditions, NRII was knocked out and NRI was recruited as a dynamic controller to activate the expression of lycopene biosynthetic genes in response to the glycolytic flux indicator AcP in engineered *E. coli* strain, resulting in improved lycopene production[21]. This was the first example of dynamic control based on sensing of a certain metabolite. Furthermore, AcP-centered oscillators were designed with the introduction of AcP-induced LacI repressor that would block the expression of NRI from the *glnAp2* promoter[22] or Pta that converts acetyl-CoA into AcP[23]. Therefore, NRI-based genetic circuits with appropriate properties are capable to control expression of related genes in response to AcP level, and will help to maintain the intracellular AcP homeostasis. Moreover, dynamic control of metabolic flux relying on biosensor responding to certain metabolite has shown promise for not only rational allocation of intracellular resources, but also improvements in yield and titer of target product, and avoidance of metabolic burden and toxic accumulation[24].

In this study, we construct a synthetic machinery to improve carbon yield with AcP as the core, using *E. coli* as the host (Fig. 1). Firstly, SCTPK is assembled by overexpression of Xfspk and SBPase, and improved by repressing expression of competing enzymes using antisense RNA (asRNA) interference. Secondly, a series of genetic circuits activated or repressed by AcP, the product of SCTPK, are developed, forming a gene-metabolic oscillator responding to intracellular AcP level. Thirdly, the synthetic machinery increases the production and yield of some industrially relevant products including 3-hydroxypropionate (3HP), polyhydroxybutyrate (PHB) and mevalonate (MVA) when growing on glucose.

## Results

### Construction of the carbon conservation pathway SCTPK in *E. coli*

The SCTPK employs six enzymes, and all enzymes, except Xfspk and SBPase, are already present in *E. coli* (Fig. 2a). For SBPase, Shb17 from *S. cerevisiae* was characterized to catalyze the specific dephosphorylation of SBP[25], and it was reported that some fructose-1,6-biphosphatases (FBPases) with highly conserved lithium-binding domains have promiscuous activity as SBPase[26,27]. Therefore, genes encoding Shb17 and FBPases with this feature from diverse bacteria (Supplementary Fig. 2) were synthesized and expressed in *E. coli* BL21(DE3) strain, and corresponding proteins fused with an His$_6$-tag were purified (Supplementary Fig. 3). In addition, His$_6$-tagged FbaA from *E. coli*, FBP aldolase which catalyzes the condensation of E4P and dihydroxyacetone phosphate (DHAP)[28], was also purified to provide the commercially unavailable SBP for the in vitro activity assay of SBPase and Xfspk. As shown in Fig. 2b, GlpX from *Synechocystis* sp. PCC6803 (SyGlpX) presented the highest SBPase activity, probably because the SBPase activity is required for the Calvin-Benson cycle for $CO_2$ fixation in cyanobacteria[29]. When the FBPase assay results (Supplementary Fig. 4) was also taken into account, SyGlpX was selected as both SBPase and FBPase activities were required for construction of the SCTPK: SBPase provides the second irreversible driving force of this cycle, and FBPase decreases the glycolytic flux and pushes the carbon flow into the SCTPK. The Xfspk from *B. longum* was employed to produce AcP from the aldol cleavage reaction of S7P, F6P and X5P, and this protein was also purified and validated by in vitro activity assay (Fig. 2c). As Xfspk activity was lower than that of SBPase, the *xfspk* gene will be carried by a high-copy plasmid to increase its expression level, and *glpX* gene of *Synechocystis* sp. PCC6803 will be integrated into *E. coli* genome under the control of a constitutive strong promoter P$_{j23119}$ (Supplementary Fig. 5) in the following strain construction.

To assemble the SCTPK in vivo, an *E. coli* BW25113 mutant Q3952 was developed as the starting strain, in which *tktAB* and *talAB* genes, responsible for the carbon rearrangement, were knocked out to exclude the interference of NOG pathway, and genes related with byproducts acetate, lactate, ethanol and formate (*poxB, ldhA, adhE, pflB*) were deleted to eliminate carbon loss (Fig. 2a). Further, the acetyl-CoA production from pyruvate decarboxylation was blocked by deletion of *aceE* gene, resulting in growth deficiency in both modified M9 medium containing 1 g/L yeast extract and LB medium, which could be compensated by addition of exogenous potassium acetate (Fig. 2d and Supplementary Fig. 6). In contrast, the strain Q3964 generated by introduction of Xfspk and SyGlpX into Q3952 Δ*aceE* recovered from the growth incapability (Fig. 2d, f), demonstrating that the acetyl-CoA pool was replenished by the constructed SCTPK. Furthermore, the *gnd* gene was deleted, that should lead to depletion of ribose-5-phosphate (R5P), the precursor for nucleotides and histidine synthesis, in a strain deficient in carbon rearrangement. However, the *gnd* mutant grew as well as its parent strain, indicating that R5P could be generated as an intermediate of the SCTPK, which is confirmed by the fact that a *gnd* mutant with incomplete SCTPK could not grow even with the presence of potassium acetate (Fig. 2e, f). All these results demonstrated that the SCTPK functions well in vivo, and can generate acetyl-CoA in a way that bypass the carbon release process from pyruvate to acetyl-CoA.

The metabolites content in the strain integrating the SCTPK was subsequently tested and showed that the AcP concentration is 1.79-fold higher than that in the wild-type strain (Fig. 2g), indicating that our engineering strategy does increase the AcP production. In the strain Q3952 *aceE*, the intracellular pyruvate concentration was 2.34-fold higher than that of the wild-type strain, and it was significantly decreased by the overexpression of Xfspk and SyGlpX, but remaining higher than the wild strain (Fig. 2h), suggesting that further optimization is required to pull the carbon flux into the SCTPK to increase carbon yield.

### Optimization of the SCTPK for higher AcP production

In the strain with the SCTPK, it was noticed that some enzymes may shunt this pathway and therefore cause carbon loss (Fig. 3a). Among them, PfkA catalyzes the reactions of F6P to FBP, and S7P to SBP, both of which are inverse reactions of SBPase; GapA enables the production of glycerate-1,3-bisphosphate (BPG), which enters the glycolysis to produce pyruvate; GapB converts E4P into 4-phospho-D-erythronate (4PE), an inhibitor of R5P isomerase (Rpi)[30] involved in the SCTPK. So, it was planned to knock down expression of these genes using asRNA interference.

As reported previously[31], a double-strand stem structure, which is created with inverted repeats flanking the asRNA, could stabilize the secondary structure of asRNA and enhance gene silencing efficiency and duration (Supplementary Fig. 7a), and our results ulteriorly illustrated that the asRNA length and transcription level, which is affected by its promoter and gene copy number, synergistically determined the efficiency of asRNA silencing (Fig. 3b and Supplementary Fig. 7). Then, the target genes *pfkA, gapA* and *gapB* were inhibited individually with 100-nt asRNA transcribed by strong promoter P$_{j23119}$, and the AcP content was detected. As shown in Fig. 3c, when expression of *pfkA* gene was suppressed, the AcP concentration was 2.23-time and 1.24-time higher than the wild-type strain and Q3964, respectively. However, high-level asRNAs targeting *gapA* and *gapB* dramatically decreased the AcP content as well as the final cell density and specific growth rate (Fig. 3c, d). In strains with corresponding asRNAs, the transcription of *gapA, gapB* and *pkfA* was reduced to 4.9%, 6.7%, and 27.2% of that of the control strain having only double-strand stem but not asRNA sequence (Fig. 3e). These results indicated that strong repression of expression of *gapA* and *gapB* severely impaired the cell growth probably due to their irreplaceable role in glycolysis and vitamin B6 synthesis (Supplementary Fig. 8).

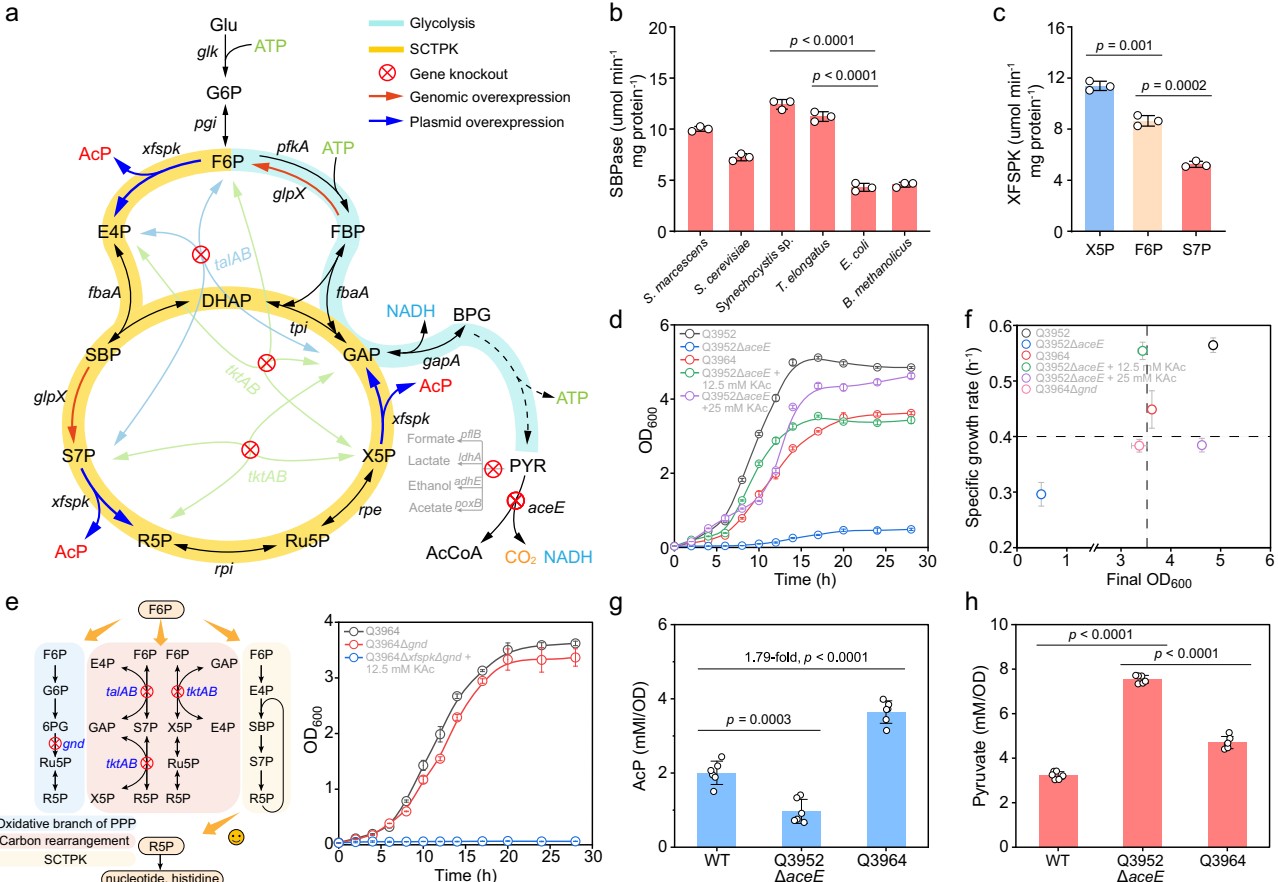

**Fig. 2 | Design and construction of the Sedoheptulose-1,7-biphosphate (SBP) Cycle with Trifunctional PhosphoKetolase (SCTPK) in *Escherichia coli*.** **a** Configuration of the initially constructed SCTPK. Phosphoketolase and sedoheptulose-1,7-bisphosphatase (SBPase) were overexpressed, genes *tktAB*, *talAB*, *pflB*, *ldhA*, *adhE*, *poxB* and *aceE* was deleted. G6P glucose-6-phosphate, F6P fructose-6-phosphate, FBP fructose-1,6-biphosphate, E4P erythrose-4-phosphate, DHAP dihydroxyacetone phosphate, S7P sedoheptulose-7-phosphate, R5P ribose-5-phosphate, Ru5P ribulose-5-phosphae, X5P xylulose-5-phosphate, GAP glyceraldehyde-3-phosphate, BPG 1,3-biphospho-glycerate, PYR pyruvate, AcCoA acetyl-CoA, AcP acetylphosphate. **b** In vitro SBPase activity of GlpX proteins from *Bacillus methanolicus*, *Saccharomyces cerevisiae* CEN.PK, *E. coli* BW25113, *Serratia marcescens*, *Synechocystis* sp. PCC6803, and *Thermosynechococcus elongatus* (n = 3 biological independent samples). **c** Phosphoketolase activity of Xfspk from *Bifidobacterium longum* with X5P, F6P, and S7P as substrate, respectively (n = 3

biological independent samples). **d** In vivo validation of the SCTPK based on the acetyl-CoA availability. The strain Q3952 (*tktAB*, *talAB*, *poxB*, *ldhA*, *adhE*, *pflB*) was used as the control. Deletion of *aceE* gene resulted in growth deficiency, which could be compensated by addition of exogenous acetate or introduction of the SCTPK (the strain Q3964 generated by introduction of Xfspk and GlpX from *Synechocystis* sp. PCC6803 into Q3952 *aceE*) (n = 3 biological independent samples). **e** A functional SCTPK rescued the growth of *E. coli gnd* mutant that is deficient in the biosynthesis of R5P (n = 3 biological independent samples). **f** The final cell density and specific growth rate of *E. coli* strains (n = 3 biological independent samples). The intracellular AcP (**g**) and pyruvate (**h**) concentration of *E. coli* BW25113, strain Q3952 Δ*aceE* and strain Q3964 (n = 6 biological independent samples). Error bar, mean ± standard deviation (SD). Two-tailed Student's *t* tests were performed to determine the statistical significance. Source data are provided as a Source Data file.

Following, combinationally regulated strains with highly inhibited *pfkA* and moderate or weak repression of *gapB* (80-nt asRNA with $P_{j23119}$ or $P_{j23118}$ promoter respectively) were constructed (Supplementary Figs. 9 and 10), and phenotypic detection showed that weak repression of *gapB* gene increased intracellular AcP concentration to 2.45 times higher than that of the wild-type strain although the final cell density and growth rate was slightly decreased (Fig. 3f, g). Then, modules mediating moderate or weak repression of *gapA* were introduced, and moderate repression of *gapA* gene increased the AcP concentration to the highest level, which was 2.83 times higher than that of the wild-type strain, and did not affect the cell growth dramatically (Fig. 3f, g). This strain with highly repressed *pfkA*, moderately repressed *gapA* and weakly repressed *gapB* was the best among all constructed strains, and was named as Q4531. As a comparison, a strain with all three genes being repressed moderately presented a bit weaker performance in both AcP production and cell growth, indicating that a hierarchical regulatory system for different genes is conducive to improve the carbon yield of the SCTPK.

Subsequently, the $CO_2$ release from wild-type *E. coli* BW25113 and Q4531 strain was determined. The $CO_2$ release rate underwent significant changes over cultivation process, and the Q4531 strain presented a maximum $CO_2$ emission rate of 4.20 mL/h, less than one-third of that of BW25113 strain (Fig. 3h). The total $CO_2$ release of Q4531 strain decreased to 47.4% compared with wild-type *E. coli* (Fig. 3i). Moreover, the strain Q4531 showed a significant decrease in pyruvate concentration relative to the wild-type strain and Q3964 (Fig. 3j), suggesting that the metabolic flow more shifted to the SCTPK following gene silencing with combinational asRNA arrays. Furthermore, the strain Q4531 showed an acetate accumulation capacity only representing 11% of the wild-type strain (Fig. 3k). So far, combinational asRNA arrays have further enhanced the effectiveness of the designed SCTPK, enhanced the production of AcP and repressed the release of $CO_2$ and accumulation of byproducts, resulting in improved carbon atom economy. In addition, it was conformed that the repeat sequences to form paired termini of asRNAs did not impair the genetic stability of corresponding plasmids (Supplementary Fig. 10).

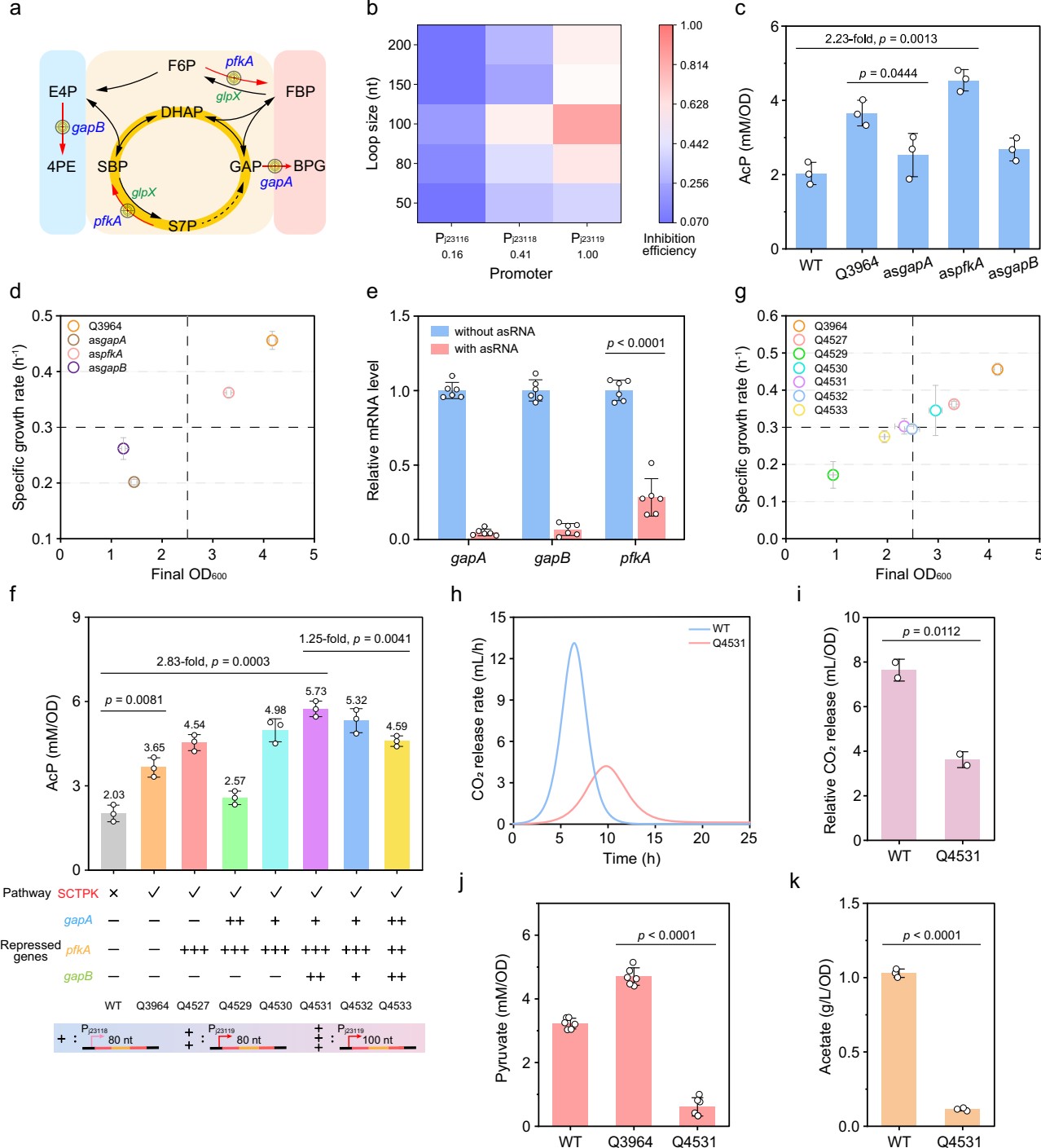

**Fig. 3 | Optimization of the SCTPK further enhanced AcP production. a** Genes that may shunt the SCTPK and will be repressed using antisense RNA (asRNA) interference. **b** Inhibition efficiency of asRNA with a paired termini was affected by the length and transcription level of the asRNA. The numbers below promoters represent the relative strengths of corresponding promoters, which were determined using GFP as a reporter and shown in Supplementary Fig. 5. **c** The intracellular AcP concentration of *E. coli* BW25113, Q3964 harboring the SCTPK, and three strains derived from Q3964 in which transcription of *gapA*, *gapB*, and *pfkA* was repressed strongly using a 100-nt asRNA transcribed from a promoter $P_{j23119}$, respectively (*n* = 3 biological independent samples). **d** The final cell density and specific growth rate of *E. coli* strains (*n* = 3 biological independent samples). **e** Relative mRNA level of *gapA*, *gapB*, and *pfkA* in strains with and without

corresponding asRNA determined by qRT-PCR (*n* = 3 biological independent samples with two technical repeats). **f** Effect of combinational and hierarchical repression of *gapA*, *gapB*, and *pfkA* on the intracellular level of AcP (*n* = 3 biological independent samples). **g** The final cell density and specific growth rate of *E. coli* strains (*n* = 3 biological independent samples). **h** $CO_2$ emission rate of *E. coli* BW25113 and Q4531 strain carrying optimized SCTPK (*n* = 2 biological independent samples). **i** Relative $CO_2$ emission of BW25113 and Q4531 (*n* = 2 biological independent samples). **j** The intracellular pyruvate concentration of *E. coli* BW25113, Q3964, and Q4531 (*n* = 6 biological independent samples). **k** Acetate accumulation in cultures of *E. coli* BW25113 and Q4531 (*n* = 3 biological independent samples). Error bar, mean ± SD. Two-tailed Student's *t* tests were performed to determine the statistical significance. Source data are provided as a Source Data file.

## Design and characterization of dynamic AcP-activated genetic circuits

As NRI protein can be directly phosphylated by AcP, it was employed here to sense AcP and modulate gene expression accordingly. Phosphorylated NRI is capable of binding to enhancer sequences upstream the promoter *glnAp2*, interacting frequently with the $\sigma^{54}$-RNA polymerase assembled at *glnAp2*, and initiating the transcription[32] (Fig. 4a). In addition, low-affinity sites for NRI-P that locates between the enhancer and the *glnAp2* promoter constitute a governor, limiting the maximum promoter activity at high concentrations of NRI-P[33].

To make this promoter AcP-inducible, the *glnL* gene was deleted, which did not affect the growth of *E. coli* obviously (Supplementary Fig. 11). In *E. coli* chromosome, the *glnLG* operon locates just downstream of the *glnA* gene, and the *glnA* terminator ($T_{glnA}$) is not effective enough to completely stop the transcription[34], resulting in a self-activating feedforward loop of NRI, which is confirmed by our results obtained from a strain carrying blue and red fluorescent protein genes flanking $T_{glnA}$ (Fig. 4b). To prevent the interference from NRI level increasing with time, a strong terminator *rrnB* T1 was introduced in front of the native $T_{glnA}$, which shut off the expression of red fluorescent protein and decreased the readthrough coefficient from 0.76 to 0.06 (Fig. 4b and Supplementary Fig. 12), suggesting the feedback loop was eliminated. These provided us a platform strain to construct the NRI-dependent AcP-responding genetic circuit.

Following, the AND-gate based circuit AI was constructed using AcP and NRI as dual inputs and green fluorescent protein (GFP) as reporter (Fig. 4c), and the dose-response of AcP was determined. The fluorescence intensity showed a tendency to increase and then decrease when AcP concentration was elevated (Fig. 4d). It was speculated that this nonmonotonic phenomenon was caused by the governor sequence which sets the upper boundary of expression from *glnAp2* promoter. Therefore, the governor was mutated to generate circuit AII, which presented a monotonic increasing response curve (Fig. 4d). To enhance the transcriptional strength from *glnAp2* promoter, *glnG* gene was overexpressed from a plasmid under the control of $P_{j23118}$ promoter instead of the genomic locus, resulting in the maximum GFP fluorescence 3.82-time higher than that of genetic circuit AII and a dynamic range of 10.4-fold (Fig. 4d, e).

To further improve the dynamic range and obtain a richer response characteristic to AcP, the interaction of NRI and enhancer was investigated comprehensively and the enhancer sequence was mutated accordingly. A helix-turn-helix (HTH) domain in the C-terminal region of *E. coli* NRI was identified to have functions of enhancer recognition and DNA binding based on the following facts: this domain is conserved in NRIs from diverse species (Supplementary Fig. 13a), it contains three conserved residues (Arg456, Asn457 and Arg461) playing essential role in affinity for DNA in *Salmonella* Typhimurium NRI[35], and *E. coli* NRI with a mutated HTH domain no longer responded to the AcP level (Supplementary Fig. 13b). Subsequently, the complex structure of DNA and *E. coli* NRI HTH domain was simulated using the binary structure of *Aquifex aeolicus* NtrC4 HTH domain and double-stranded DNA (PDB: 4FTH) as template (Fig. 4f), and the result suggested that a DNA sequence GGTGCA formed the most stable complex with those three conserved amino acid residues through hydrogen bond and electrostatic force (Fig. 4g). Based on this structure, three enhancer sequences with different affinity for NRI were designed to replace the native enhancer sites 1 and 2 of *glnAp2* promoter in genetic circuit AIII (Fig. 4g, h), resulting in a small library containing nine mutated *glnAp2* promoters. All these biosensors still showed a response to AcP concentration, and it seemed that the enhancer's affinity for NRI corelated positively with the strength of *glnAp2* promoter i.e., the fluorescence intensity at a certain AcP concentration, but corelated negatively with the dynamic range of corresponding biosensors (Fig. 4i). Taken together, the circuit AIIIf was selected with the highest dynamic range of 30.3-fold, low basal leakage

expression and acceptable expression strength at high AcP level (Fig. 4d).

Next, the expression of NRI was adjusted. When the *glnG* gene was overexpressed under control of a weaker promoter $P_{j23116}$, the dynamic range increased to 57.1-fold, however, the maximum output signal was lower than that of the circuit AIIIf (Fig. 4d, e). In the case that the *glnG* gene was expressed from its original genomic locus, the maximum output signal dropped further, and the dynamic range was 27.9-fold (Fig. 4d, e). Collectively, a series of AcP-activated genetic circuits with different switching time, regulatory intensities and dynamic ranges were obtained by adjusting the cis-acting sequences (enhancers and governors) and trans-acting element NRI expression of the promoter *glnAp2*.

## Design and characterization of dynamic AcP-inhibited genetic circuits

The AcP negative-response genetic circuit was constructed based on the following idea, that is, the genomic promoter *glnAp2* with mutated governor sequence is used to modulate the expression of a transcriptional repressor PhlF from *Pseudomonas fluorescens*[36], which will repress expression of the reporter gene *gfp* from the protomer $P_{phlF}$ (Fig. 5a). In the strain carrying an initial construct RI, the PhlF protein was expressed specifically in response of AcP addition (Fig. 5b). Moreover, AcP-dependent repression of GFP fluorescence was observed, and this repression was undermined by addition of 2,4-deacetylphloroglucinol (DAPG) which can bind to and release PhlF protein from $P_{phlF}$ promoter (Fig. 5c), suggesting that the repression was mediated by the PhlF repressor but not a direct effect of AcP. However, this strain presented high response threshold of AcP and quantitative leaky expression at high AcP concentration.

To improve this repression configuration, an extra copy of PhlF binding site *phlO* was inserted into the $P_{phlF}$ promoter to strengthen the affinity of PhlF repressor for this promoter. Between the −35 and −10 regions of the $P_{phlF}$ promoter, a 24-bp sequence containing inverted repeats was regarded as putative *phlO* operator, and its mutation totally abolished the PhlF-mediated repression on the $P_{phlF}$ promoter (Fig. 5c). Following, one and two copies of *phlO* operator were insert downstream of the transcription start site respectively as shown in Fig. 5d. Surprisingly, a stepwise decrease in fluorescence was observed even with the absence of PhlF protein after the insertion of *phlO*s, but the *gfp* mRNA level was not significantly changed (Fig. 5d). It was believed that this phenomenon was caused by the secondary structure of 5′-untranslated region which was stabilized by additional *phlO* sequence (Supplementary Fig. 14) and affected the translation initiation of downstream genes. Then, a *phlO* operator was inserted upstream the $P_{phlF}$ promoter with the distance between the centers of two *phlO* sequences of 63-bp. However, this insertion didn't change the expression pattern of GFP with the presence and absence of PhlF protein neither (RIII in Fig. 5e and Supplementary Fig. 15).

To make these operators acting synergistically, one method is generation of DNA looping by a protein or protein complex that simultaneously binds to two detached sites on a DNA molecule[37]. As PhlF protein usually forms a dimer binding to two inverted repeats in a *phlO* operator and cannot form higher-order complex[36], it was fused with C-terminal domain of the λ repressor cI[38] which mediates an intermolecular interaction resulting in the cooperative binding of two PhlF dimers to adjacent *phlO* operators (Fig. 5e). As we had expected, introduction of the oligomerization domain reduced the response threshold of AcP and leaky expression effectively. Subsequently, several peptide linkers were inserted between PhlF and cI domain, respectively, to increase spatial separation and to avoid the mutual interference between domains. Among them, the linker (PT)4 P showed the highest execution of suppression (Fig. 5e).

Next, the distance between two *phlO* operators was adjusted to explore its effect on DNA looping. As shown in Fig. 5f, the fluorescence

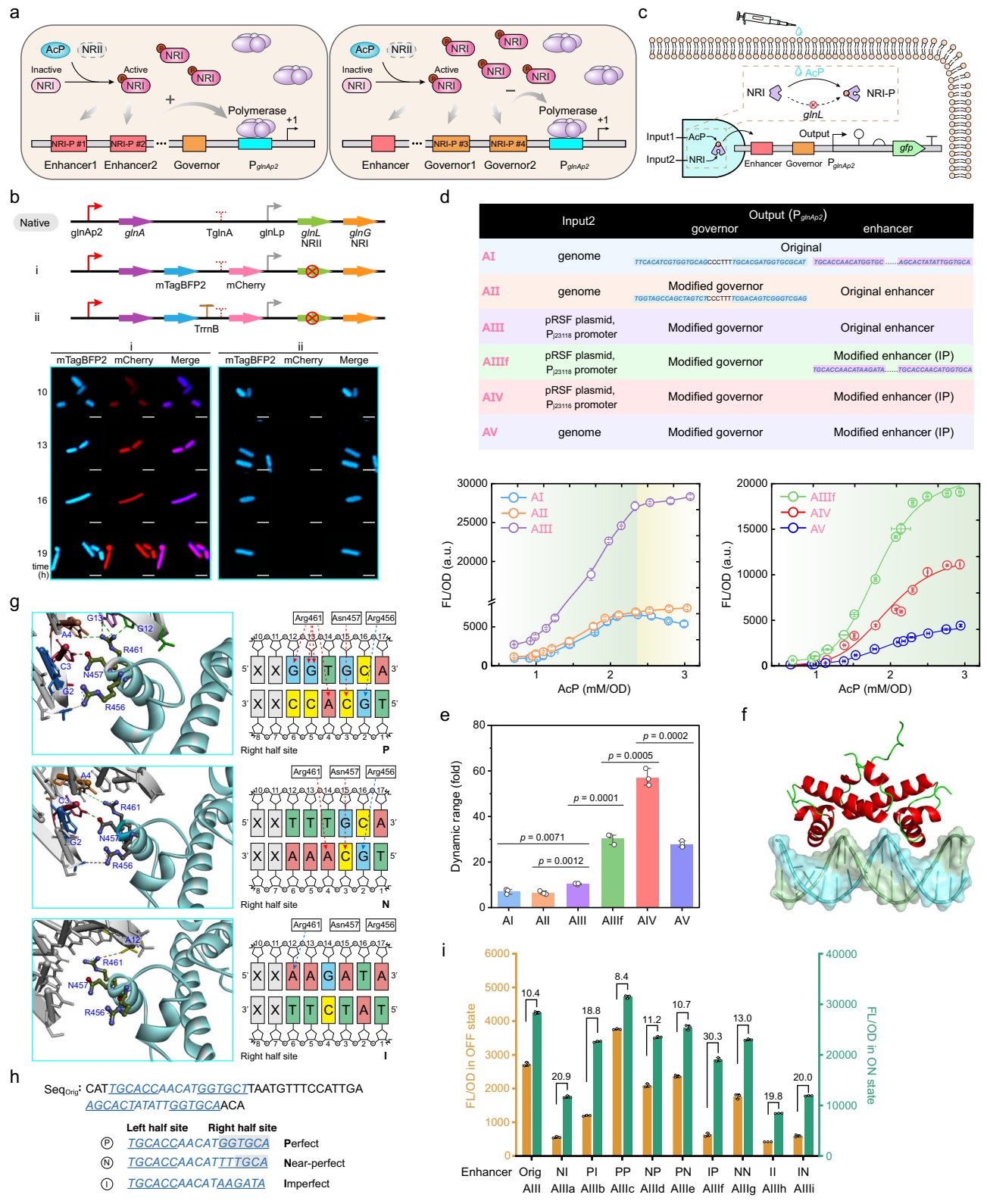

output increased as the *phlO*ᵤₚ was gradually moved away, and the DNA looping-dependent repression mainly functioned in the distance range of 47–297 bp. Moreover, the spacer was further refined to 48, 50, 52...78, 80 bp, and the output signal showed a distance-related periodic fluctuation with a period of 11-bp, which is exactly the number of base pairs per helical turn of double-stranded DNA (Fig. 5g). These results suggested that formation of DNA looping is affected by the relative direction of two operators on DNA molecule. Specifically,

when the two operators are positionally deflected or located on opposite sides, the DNA looping will be structurally unstable due to the rigid action of DNA helix, resulting in a decrease in its repression performance. In addition, it was proved that the space packing formed between *phlO*ᵤₚ and P*phlF* promoter was neglectable and does not affect the transcription from P*phlF* promoter as long as their distance was over 7 bp (Supplementary Fig. 16). Furthermore, the AcP response properties of three genetic circuits with engineered PhlF repressor and

**Fig. 4 | Construction and characteristics of NRI-based AcP-activated genetic circuits. a** The regulatory mechanism of two-component system Ntr on promoter *glnAp2*. The regulator NRI (encoded by *glnG*) can be phosphorylated by the sensor kinase NRII (encoded by *glnL*) or AcP, bind to the enhancer sequence and activate the *glnAp2* promoter, however excessive phosphorylated NRI will bind to the governor and limit the maximum promoter activity. **b** Insertion of a strong termi-nator *rrnB* T1 eliminated the self-activating feedforward loop of NRI. Genes encoding blue and red fluorescent proteins were inserted into *E. coli* chromosome flanking *glnA* terminator, and cells were observed using laser confocal microscopy. The scale bars were 2 μm. Similar results were obtained from three biological independent samples, and a representative result was displayed. **c** Model illus-trating a tunable AND gate formed by AcP and NRI as inputs and GFP expression as output. **d** The structural and responsive characteristics of AcP-activated circuits (*n* = 3 biological independent samples). pRSF plasmid is a derivate of pRSFDuet-1, in which the *lacI* gene and T7 promoter were deleted. **e** Dynamic range of the AcP-

activated genetic circuits (*n* = 3 biological independent samples). **f** Docking view of the helix-turn-helix (HTH) region of *E. coli* NRI bound to enhancer. **g** Simulation of interactions of *E. coli* NRI HTH region with the right half-sites of three artificially designed enhancers P, N, and I. Dashed red and blue lines represent hydrogen bonds and electrostatic force between NRI and enhancers. **h** Mutated enhancer sequences of *glnAp2* promoter. Enhancers containing perfect (P), near-perfect (N), and imperfect (I) palindrome sequences were used to replace the original enhance 1 and 2 to generate nine mutated *glnAp2* promoters. **i** Dynamic range of AcP-activated circuits with mutated enhancer sequences in *glnAp2* promoter (*n* = 3 biological independent samples). "Orig" represents the original enhancer sequence, while the rest refer to different combinations of P, N, and I sequences at enhancer sites 1 and 2. Error bar, mean ± SD. Two-tailed Student's *t* tests were performed to determine the statistical significance. Source data are provided as a Source Data file.

different distance of two *phlO* operators of 60, 63, and 66 bp, respectively (Fig. 5h) were confirmed. The results showed that all three genetic circuits presented significantly improved repression profile than the initial construct, and the one with 66-bp spacer between *phlO* operators has faster signal response, better cooperative ability, and lower leakage expression than the others.

Finally, the portability of the dynamic AcP-activated and -inhibited genetic circuits were verified by replacing the reporter GFP with β-galactosidase (β-gal), and appropriate expression profile of β-gal was obtained under conditions with various intracellular AcP concentra-tions (Supplementary Fig. 17). In summary, a series of AcP responsive genetic circuits with different properties were constructed, and can be used to control expression of metabolic enzymes.

## Establishment of AcP-centered oscillation system

As dynamic expression of metabolic genes is able to better manage the trade-offs between growth and production and can enhance the yield and titer of target products[24], the AcP-responding circuits will be employed to control the transcription of genes related in AcP produc-cing and consuming routes. In specific, the AcP-activated circuits are used to control the AcP consumption route, that is, the acetyl-CoA production catalyzed by phosphate acetyltransferase (Pta), and the AcP-inhibited circuits are used to control the AcP generation catalyzed by Xfspk in SCTPK. As a consequence, bacterial fate is going to be divided into two modes of mutual cycling (Fig. 6a). The manifestation and alternation of these two fates will be determined only by the intracellular concentration of AcP.

To implement this scheme, the constructed activating circuit AI was used to control transcription of genes *pta* and *mcherry*, as well as circuit RI to control expression of *xfspk* and *gfp*, to generate a proto-type device I (Supplementary Fig. 18). As expected, the fluorescence expression curves suggested that GFP and mCherry exhibited oscilla-tory expression in opposite phases. However, a higher amplitude of GFP than mCherry was observed and the oscillatory expression gra-dually became blurred after two cycles (Fig. 6b). In oscillator II, the *glnAp2* promoter was replaced by its derivative with optimized enhancer sequence, resulting in a reduced expression leakage of mCherry, but the oscillatory behavior was still unsustainable (Fig. 6c).

Next, in oscillator III, the activator NRI was expressed from a plasmid under control of P$_{j23116}$ promotor instead of its genomic locus to increase the NRI level, and the deterrent driver of the repression branch was switched to engineered PhlF-(PT)4P-*c*I and P$_{phlF}$ promoter with double *phlO* operators to improve the repression efficiency. As a consequence, a sustained oscillatory state with a cycle of about 12 h was observed, and two modules had similar amplitudes and opposite phases, demonstrating that this device had a better synchronization tunability, higher robustness, and stronger resistance to noise (Fig. 6d). Further increase of NRI level by using a stronger promoter P$_{j23118}$ in oscillator IV accelerated the oscillatory rhythm, resulting in

the switch of two modes in every 9 h (Fig. 6e). However, there was a certain degree of leakage expression of *mCherry* gene in the activation module, and the amplitude of mCherry was also higher than that of GFP. To lower the background expression of mCherry, the *lacI* gene was placed downstream of the P$_{phlF}$ promoter in the repression module to produce the repressor LacI negatively correlating with intracellular AcP level which will block the *glnAp2* promotor in the activation module. Interestingly, this not only reduced the mCherry expression and prolonged the switchover time compared with Oscil-lator IV, but also lead to gradually decayed amplitude of mCherry and GFP to generate a damped oscillation (Fig. 6f). Furthermore, color profiles with multiple visual gradients were observed in *E. coli* cells carrying the synthetic oscillatory networks (Supplementary Fig. 19), indicating the effectiveness and diverse responsive features of these oscillatory circuits.

To directly evaluate the effect of oscillation system, intracellular AcP concentration and transcriptional level of *pta* and *xfspk* genes were determined in the strain Q4602 carrying SCTPK and Oscillator III. As shown in Fig. 6g, the AcP level fluctuated around 2 mM/OD, which was much lower than that in the strain Q4531 carrying SCTPK alone. Furthermore, decrease of the AcP concentration also brought the protein acetylation back to the level similar to that of the wild-type strain (Fig. 6h), avoiding possible unfavorable regulatory effects caused by AcP accumulation. In addition, the genes *pta* and *xfspk* were transcribed oscillatory in opposite phases, and the mRNA level of *pta* changed synchronously with intracellular AcP concentra-tion (Fig. 6i).

Besides the periodic oscillation over time, our oscillator could also enable spatial patterning. As shown in Fig. 6j, the strain Q4602 carrying SCTPK and Oscillator III was spread homogeneously onto an agar plate, and a central source of AcP was added to form a radial gradient. By expressing different fluorescence protein, the isogenic bacterial population interpreted the AcP gradient into two discrete color zones: red and green, just like the Bangladesh flag. In contrast, a bacterial population carrying unregulated *gfp* and *mCherry* genes only presented a uniform yellow in spite of the AcP gradient.

Overall, five oscillating devices were constructed through the integration of transcriptional regulation and engineered metabolism, which could be classified into three categories: unsustained, sustained, and damped oscillation. The constructed oscillatory network is cap-able of regulating expression of metabolic genes precisely, sponta-neously, and real-time to maintain the homeostasis of AcP in *E. coli* cells.

## Application of AcP-centered oscillation system in biosynthesis

The ultimate goal of constructing artificial network including SCTPK and oscillators is to improve the carbon atom economy during bio-synthesis of chemicals and materials. In this network, the SCTPK is the "AcP generation" part, channeling more carbon atoms to central

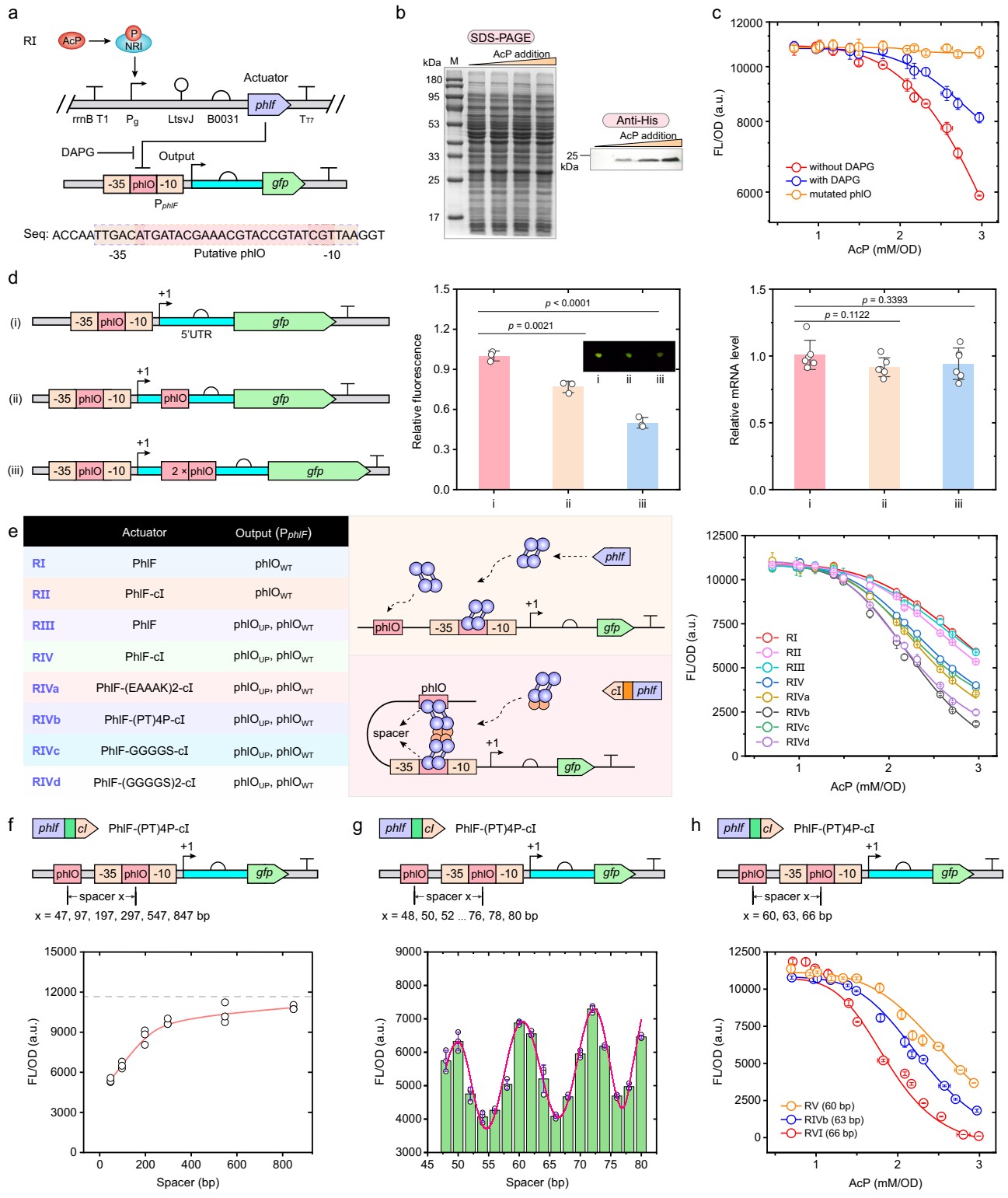

metabolic intermediate AcP and increasing the theoretical upper limit of atomic conversion, biosynthetic pathway genes along with *pta* compose the downstream "bioproduction" section, and the oscillator regulates transcription of related genes in response to the intracellular AcP level (Fig. 7a). Here, three products, 3HP, PHB and MVA, were adapted as proxy for proof of concept (Fig. 7b). To test whether microbial cell factory equipped with this oscillator system possess better performance, two kinds of strains in which the biosynthetic genes are regulated with static control model and dynamic switch were also constructed (Supplementary Fig. 20).

3HP was ranked in the top 12 value-added chemicals from biomass[39], and is produced from acetyl-CoA by the acetyl-CoA carboxylase ACC and malonyl-CoA reductase MCR[40]. MCR catalyzes the two-step reduction of malonyl-CoA to 3HP, and its N- and C-terminal fragments are functionally distinct, showing alcohol dehydrogenase and aldehyde dehydrogenase activity, respectively. Furthermore, segmental expression of MCR led to higher overall enzyme activity and improved 3HP production[7,41]. So dissected genes *mcr-n* and *mcr-c* were used here. Under static control model, the genes *xfspk*, *pta*, *accABCD*, *mcr-n* and *mcr-c* were expressed constitutively when required in wild-

**Fig. 5 | Construction and characteristics of AcP-inhibited genetic circuits.**
**a** Design of the AcP negative-response circuit. *Pseudomonas fluorescens* repressor gene *phlF* was controlled by *glnAp2* promoter, and PhlF protein will inhibit expression from promoter P*phlF*. Putative PhlF binding sequence was shown. 2,4-Deacetylphlorogluicnol (DAPG) can release PhlF from P*phlF* promoter. **b** Expression of PhlF protein at different AcP concentrations. The cell lysates of strain Q4562 carrying the AcP-inhibited circuit RI grown under various AcP concentrations were separated by SDS-PAGE, and visualized by Coomassie blue stain and anti-His$_6$ immunoblot. Similar results were obtained from two biological independent samples, and a representative result was displayed. **c** Validation of the PhlF-dependent AcP-inhibited circuit (*n* = 3 biological independent samples). The repression was undermined by *phlO* sequence mutation and DAPG addition. **d** Effect of additional *phlO* operators downstream of P*phlF* promoter on AcP-inhibited circuit. GFP fluorescence (*n* = 3 biological independent samples) and mRNA level (*n* = 3 biological independent samples with two technical repeats) with the absence of PhlF protein

were shown. **e** Effect of additional upstream *phlO* operator and PhlF oligomerization on AcP-inhibited circuits (*n* = 3 biological independent samples). A *phlO* operator was inserted upstream of P*phlF* promoter with the distance of the centers of two *phlO* sequences of 63-bp. C-terminal domain of the λ phage repressor cI and different linker peptides were fused with PhlF to mediate its oligomerization. **f** Effect of the distance between two operators on AcP-inhibited circuits (*n* = 3 biological independent samples). PhlF-(PT)4P-cI protein was used. The dotted line represented the fluorescence intensity without upstream operator. **g** Further refining the influence of the distance between two operators (*n* = 3 biological independent samples). Genetic circuits with the distance of 48-80 bp were constructed and tested at the AcP concentration of 2.14 mM/OD. **h** Characteristics of AcP-inhibited circuits with engineered PhlF and operators (*n* = 3 biological independent samples). Error bar, mean ± SD. Two-tailed Student's *t* tests were performed to determine the statistical significance. Source data are provided as a Source Data file.

type *E. coli* BW25113, Q3952, and Q4531. The strains with dynamic switch model expressed *xfspk* gene constitutively, and transcription of other genes was activated by AcP sensors AI and AV. In contrast, these genes were subjected to automatic dynamic activation or repression using the oscillators I-IV in other strains (Supplementary Fig. 21). In shaking flask cultivation, the integration of SCTPK and oscillators slightly slowed down the growth and reduced the final cell density (Fig. 7c), whereas significantly enhanced the 3HP production and yield (Fig. 7d). The strain Q4607 (BW25113 carrying 3HP pathway) accumulated 0.93 g/L 3HP with a yield of 0.09 g/g glucose after 72-h cultivation, deletion of genes in the strain Q4634 (Q3952 carrying 3HP pathway) slightly enhanced the 3HP production, while integration of SCTPK increased the 3HP titer and yield to 1.75 g/L and 0.23 g/g glucose, respectively. Following, the AcP-dependent dynamic switch and oscillators further improved the bioproduction of 3HP. Finally, the strain Q4617 harboring SCTPK and oscillator IV presented the highest 3HP titer of 7.60 g/L and yield of 0.71 g/g glucose, which were both more than 8 times higher than those of the original strain Q4607 (Fig. 7d). Meanwhile, the strain Q4607 accumulated a considerable amount of acetate and lactate, 4.67 g/L and 3.07 g/L respectively, and only 0.33 g/L acetate was detected in culture of strain Q4617 (Fig. 7e) probably due to the decreased intracellular pyruvate level. This result suggested that the SCTPK and oscillator improved the carbon atom economy during 3HP fermentation in two ways, enhancement of acetyl-CoA yield from glucose and inhibition of by-products synthesis.

Furthermore, fed-batch fermentation was carried out in a 5-L bioreactor using the strain Q4617. During fermentation, transcription of *pta* and *xfspk* fluctuated periodically in response to the intracellular AcP concentration (Fig. 7g, h), suggesting that our oscillator system has high robustness even in a larger scale cultivation and is able to sense AcP and regulate expression of metabolic genes precisely and persistently. It was also determined that the aeration ratio played an important role in 3HP production (Supplementary Fig. 22). Under optimized conditions, our recombinant strain Q4617 produced 73.42 g/L 3HP with a yield of 0.51 g/g glucose after 108-h fermentation (Fig. 7f). To the best of our knowledge, this is the highest 3HP production reported via the malonyl-CoA pathway in engineered microbes.

MVA is a valuable precursor for terpenoid biosynthesis, and PHB is a biodegradable and biocompatible thermoplastic. Subsequently, the synthetic genes of MVA and PHB were assembled into *E. coli* recombinant strains with three different regulation models, respectively (Supplementary Figs. 23 and 24), and fermentation in shaking flask and 5-L bioreactor was performed (Fig. 7i, j and Supplementary Figs. 25 and 26). Similar with 3HP, reconfiguration of central metabolism to the SCTPK increased MVA production from 0.78 g/L to 1.25 g/L and PHB production from 0.42 g/L to 1.00 g/L, and the yields of MVA and PHB from glucose were increased by 1.29- and 1.06-fold, respectively. When the AcP-dependent dynamic

switch and oscillators were assembled, the bioproduction of MVA and PHB were further improved. The strain Q4624 presented the highest MVA titer, 4.67 g/L which was 5.99 times higher than that of the original strain Q4618, and the MVA yield was 0.61 g/g glucose, reaching 95% of the theoretical yield of MVA using SCTPK (0.64 g/g) and exceeding its native theoretical yield (0.54 g/g). The best strain for PHB production was Q4632 which accumulated 3.31 g/L PHB, and its conversion efficiency of glucose to PHB has attained 0.41 g/g, reaching 85% and 71% of its native theoretical yield (0.48 g/g) and theoretical yield using SCTPK (0.58 g/g) respectively. In addition, excretion of byproducts acetate and lactate was suppressed during the fermentation of MVA and PHB (Supplementary Fig. 25c, d). All these results demonstrated that we have developed a universal *E. coli* platform to produce acetyl-CoA derived chemicals and materials with high carbon atom economy.

## Discussion

In this study, a universal synthetic machinery with high carbon atom economy was constructed. Firstly, the carbon conserving pathway SCTPK, converting glucose into stoichiometric amounts of AcP without carbon loss, was established relying on the introduction of trifunctional phosphoketolase and SBPase (Fig. 1). Secondly, oscillatory systems were developed using genetic circuits with various responsive properties to AcP, maintaining intracellular AcP homeostasis and balancing the cellular resources between cell growth and the synthesis of target chemicals. Finally, biosynthesis of several chemicals was carried out using this synthetic machinery, representing much higher production and yield of each product when compared with natural central metabolism of *E. coli* or flux regulation modes including commonly used static control and dynamic switch. Our results demonstrate that the combination of construction of carbon conserving pathway and dynamic oscillatory regulation in response to a key metabolite provides an attractive strategy for improving the performance of microbial cell factories.

For heterotrophic microorganism, carbon yield is one of the most important parameters for evaluating their industrial biomanufacturing performance[42]. A typical bioproduction system loses more than 30% of carbon during the production process. There are two main approaches to raise the upper bound of carbon yield. One is to employ carboxylation reactions to fix $CO_2$, like the succinate biosynthetic pathway in which succinate was derived from acetyl-CoA with two $CO_2$ fixation reactions[43], but this only works with specific metabolic pathways. The other approach is to avoid decarboxylation, as NOG does in bypassing pyruvate decarboxylation to eventually generate acetyl-CoA[12], a cornerstone of a variety of metabolites including lipids, alkanes and polymers.

Here, another carbon conserving pathway SCTPK was demonstrated to be efficient and manipulable in *E. coli*. This trifunctional phosphoketolase-dependent cycle was constructed and optimized

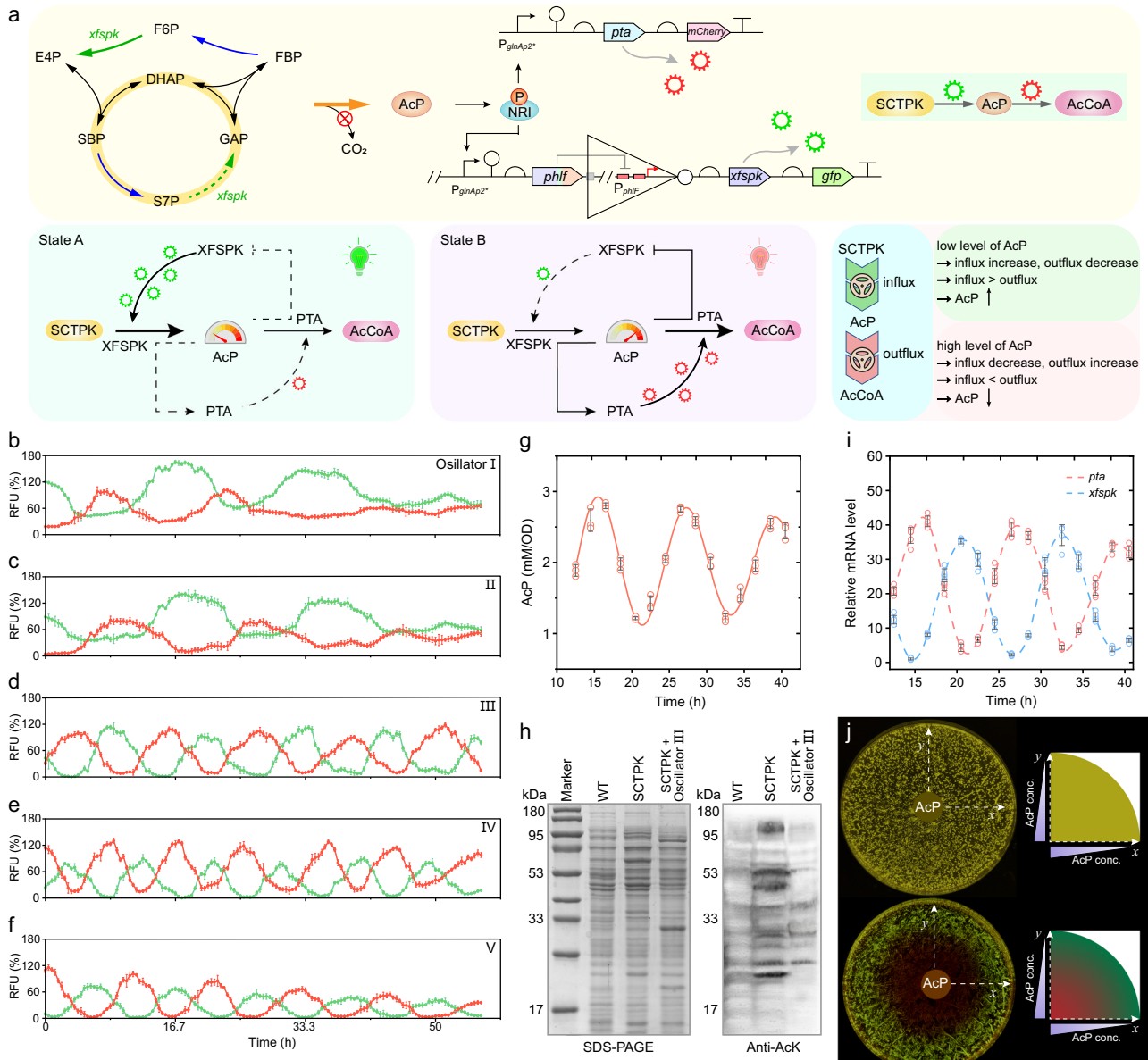

**Fig. 6 | Design and verification of the AcP-centered oscillation system. a** The alternating cellular states presented by AcP-response oscillators. This metabolator consists of a network with pools of two metabolite AcP and acetyl-CoA produced by enzymes Xfspk and Pta, whose expressions are negatively and positively regulated by AcP, respectively. In State A where the AcP is low, Xfspk is expressed along with GFP. The accumulation of AcP represses Xfspk and upregulates Pta and mCherry, generating acetyl-CoA from AcP. With the decrease of AcP level and increase of acetyl-CoA level, Xfspk is expressed again and Pta is deactivated, returning to State A. **b**–**f** Quantification of the relative fluorescence over time of *E. coli* strains carrying Oscillators I-V, respectively (*n* = 6 biological independent samples). Detailed structures of these oscillators were shown in Supplementary Fig. 18. **g** The intracellular AcP concentration of *E. coli* strain Q4602 carrying the SCTPK and Oscillator III (*n* = 3 biological independent samples). **h** Anti-acetyllysine immunoblot analyses of whole cell proteins of *E. coli* BW25113, Q4531 carrying the SCTPK, and Q4602. The whole-cell lysates were separated by SDS-PAGE, and visualized by Coomassie blue stain and anti-acetyllysine (anti-AcK) immunoblot. Similar results were obtained from two biological independent samples, and a representative result was displayed. **i** Relative mRNA level of *xfspk* and *pta* in strain Q4602 determined by qRT-PCR (*n* = 3 biological independent samples with two technical repeats). **j** Spatial pattern formed by strain Q4602. Cells were spread homogeneously onto agar plate, and a paper disk soaked with 15 μL of 1 M AcP was placed in the center to create a radial AcP gradient. A strain with constitutively expressed GFP and mCherry (upper section) was used as control. Error bar, mean ± SD. Two-tailed Student's *t* tests were performed to determine the statistical significance. Source data are provided as a Source Data file.

holistically using the pull-push-promote strategy. Specifically, irreversible reactions catalyzed by Xfspk and SBPase provide driving force and push carbon flux into this carbon conserving cycle (Fig. 2a). Secondly, expression of gene *xfspk* encoding rate-limiting enzyme was strengthened by placing this gene on a high copy number plasmid, to promote synthesis of AcP from S7P, F6P and X5P. Thirdly, genes involved in production of acetate, lactate, ethanol and formate were knocked out to eliminate carbon loss as a way to pull byproduct

carbon into the target pathway, and expression of PfkA, GapA, and GapB proteins that may shunt the SCTPK was repressed hierarchically by combinational asRNA arrays to further improve the function of SCTPK (Figs. 2a and 3a). This engineering strategy has achieved excellent results: the resultant strain Q4531 presented an intracellular AcP concentration of 5.73 mM/OD, 2.83 times higher than that of *E. coli* BW25113 wild-type strain (Fig. 3f), and the maximum $CO_2$ release rate of Q4531 strain only accounted for less than one-third of that of the

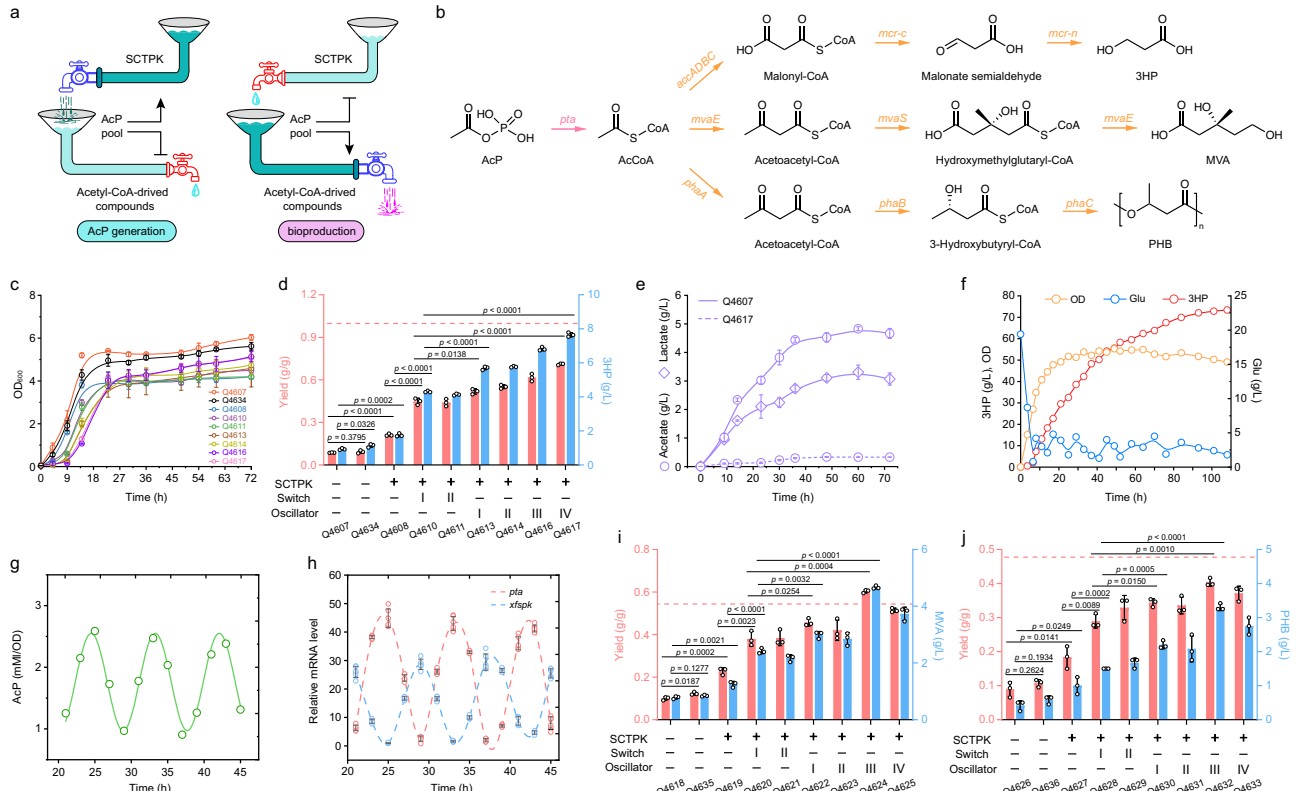

**Fig. 7 | Application of AcP-centered oscillation system in the production of 3-hydroxypropionate (3HP), mevalonate (MVA), and polyhydroxybutyrate (PHB). a** Schematic of the artificial network divided into two models for biosynthesis of chemicals and materials. **b** Biosynthetic pathways of 3HP, MVA, and PHB using AcP as precursor. Genes were regulated under three different strategies including static control, dynamic switch, and oscillatory regulation as shown in Supplementary Figs. 21, 23–24. *pta*, phosphate acetyltransferase; *mvaE*, acetyl-CoA acetyltransferase/HMG-CoA reductase; *mvaS*, hydroxymethylglutaryl-CoA synthase; *phaA*, acetyl-CoA acetyltransferase; *phaB*, 3-hydroxybutyryl-CoA dehydrogenase; *phaC*, 3-hydroxybutyrate polymerase; *accADBC*, acetyl-CoA carboxylase complex; *mcr*, malonyl-CoA reductase. **c** Growth curve of strains for 3HP production during shaking flask cultivation using glucose as sole carbon source (*n* = 3 biological independent samples). **d** The 3HP titer and yield of strains

with various regulatory strategies (*n* = 3 biological independent samples). The dashed line represents the native theoretical yield of 3HP from glucose. **e** Accumulation of acetate and lactate of strains Q4607 and Q4617 in 3HP production (*n* = 3 biological independent samples). The lactate was not detected in Q4617 culture. Cell growth and 3HP production (**f**), intracellular AcP concentration (**g**) and relative mRNA level of *xfspk* and *pta* (**h**) of strain Q4617 during an aerobic fed-batch fermentation in a 5-L bioreactor. Similar results were obtained from two biological independent samples, and a representative result was displayed. Production of MVA (**i**) and PHB (**j**) using strains with various regulatory strategies in shaking flask cultivation (*n* = 3 biological independent samples). The dashed line represents the native theoretical yield. Error bar, mean ± SD. Two-tailed Student's *t* tests were performed to determine the statistical significance. Source data are provided as a Source Data file.

BW25113 strain (Fig. 3h). When coupled with MVA biosynthetic pathway and AcP-responsive oscillator, the MVA yield from glucose reached 0.61 g/g, exceeding the native MVA theoretical yield of 0.54 g/g (Fig. 7i). All these results demonstrated that SCTPK did enhance the carbon atom economy during bioconversion process, and it could be adopted as a new carbon sequestration platform to produce more critical or value-added chemicals.

However, there is still an issue, that is, the SCTPK alone does not generate ATP and reducing power (Fig. 2a), which are required for cell growth and production of chemicals such as 3HP, MVA and PHB. In our strains, *E. coli* native pathways for glucose catabolism including glycolysis, Entner-Doudoroff pahway and residual upper section of phosphate pentose pathway, are still functional, and can produce energy and reducing equivalents. Furthermore, cells can generate ATP and NAD(P)H via the TCA cycle and respiration. Therefore, we believe that ATP and reducing power required for cell growth and biosynthesis are supplied by *E. coli* native pathways, and main role of SCTPK is to produce AcP from glucose without $CO_2$ release to improve the production and yield of acetyl-CoA-derived chemicals. This situation is similar to recombinant strain carrying NOG reported previously[44]. Currently, there are several hybrid routes being established by simultaneous overexpression of NOG and basal central metabolism[45–47].

These strategies essentially replenish energy with carbon atom, and this stitching practice results in a theoretical carbon yield of 61.1-83.3%[48]. Now electrical power can be produced in sustainable ways, and can be used to supply reducing equivalent and energy in microbial electrosynthesis system[49]. Application of electrical driven generation of ATP and reducing equivalent in carbon conserving platform strains is believed to be an effective solution.

In this study, we designed and constructed several oscillator systems that response to AcP spontaneously. They sense the balance between two inter-converting metabolite pools (AcP and acetyl-CoA) produced by two enzymes Xfspk and Pta, whose expressions are negatively and positively regulated by AcP, respectively (Fig. 5). Compared with the previously constructed NRI-dependent AcP sensors[22,23], our genetic circuits were optimized by adjusting the affinity of activator NRI or repressor PhlF to corresponding promoters, resulting in more diverse oscillator systems with longer duration which have a wide range of applications in biosynthesis of various chemicals and materials. For example, the highest productions of MVA and PHB were achieved with the usage of Oscillator III while 3HP with Oscillator IV (Fig. 7d, i, j). In strains carrying the sustained oscillators, periodic fluctuations of intracellular AcP concentration and gene expression level were observed in both shaking flask cultivation and fed-batch

fermentation (Figs. 6g, i and 7g, h), suggesting great process robustness and scalability of these oscillation systems. The metabolic networks of microbials are highly regulated and respond to environmental conditions. This adaptability may benefit cell survival and growth, while it often causes inconsistent fermentation performance and impairs the product synthesis in an industrial process, resulting in difficulty in process development and scaling[50]. It is believed that these oscillation systems, which makes biological process more predictable and repeatable, have potential to simplify the optimization and scale up of bioproduction of other valuable chemicals.

Oscillatory regulation of carbon flux significantly improved the production and yield of target chemicals and materials in our study. In metabolic engineering, there are several strategies to modulate the metabolic flux. For example, static regulation means are easy to operate and have short cycle time, such as overexpression and knockout of genes, modifications of RBS sites, optimization of plasmid copy number. Nevertheless, these approaches usually bring issues like intermediate metabolite accumulation and carbon source waste[9,51]. Dynamic switch alleviates the effect of exogenous pathway introduction on the uneven distribution of resources within the cell. However, when there is excessive accumulation or consumption of a key intracellular component or compound, it would affect the global metabolism of the cell[9,51,52]. Here oscillatory model based on sophisticated sensing of key metabolite AcP, the immediate product of the SCTPK and precursor of acetyl-CoA and biosynthesis, provides an alternative in the face of the above problem. Our oscillators can balance the flux of AcP producing and consuming routes autonomously, maintain the intracellular AcP homeostasis (Figs. 6g and 7g), and avoid excessive protein acetylation (Fig. 6h), resulting in incrasing of carbon yield in bioproduction. Compared with E. coli strains with statically reconfigured central metabolism network, utilization of the AcP-centered oscillators improved the productions of 3HP, MVA, and PHB by 4.3, 3.7, and 3.3 times, respectively, in shaking flask cultivation (Fig. 7d, i, j). In fed-batch fermentation, 3HP was accumulated up to 73.42 g/L with a yield of 0.51 g/g glucose (Fig. 7f), representing the highest 3HP production and yield reported via the malonyl-CoA pathway.

## Methods

### Strains and plasmids

All plasmids and strains used in this study are listed in Supplementary Data 2 and 3, respectively. The primers and the sequences of genetic parts used in this study are listed in Supplementary Data 4 and 5, respectively. E. coli strain DH5α was used for plasmid construction. E. coli BL21(DE3) and plasmid pET28a were used for overexpression of FbaA, Xfspk and SBPases from six different strains, which were further used for protein purification. E. coli BW25113 was used as the host for SCTPK construction, optimization, and application. Two artificially constructed plasmids based on pETDuet-1 and pRSFDuet-1 were used to express related enzymes in the product synthesis pathway. Specifically, pETD was used for combinatorial expression of Xfspk and asRNA arrays, while pRSF was used for combinatorial expression of PTA and enzymes downstream of specific pathway (MvaE and MvaS for mevalonate[53]; AccADBC, MCR-N and MCR-C for 3HP[7]; PhaC, PhaA and PhaB for PHB[54]; sequences shown in Supplementary Data 6). All constructs used for tools development and application were generated using standard restriction cloning with enzymes from Takara Bio (Dalian, China) or ClonExpress MultiS One Step Cloning Kit procedure from Vazyme Biotech (Nanjing, China). PCR purification and miniprep kits (Omega, USA) were used for DNA extraction following the manufacturer's protocols. Competent cells were prepared by Ultra-Competent Cell Preps Kit (Sangon Biotech, China). Primers were purchased from a commercial synthesis service (Tsingke, China). All constructed plasmids were confirmed by DNA sequencing (Tsingke, China).

For the construction of SCTPK, rock-paper-scissors based CRISPR/Cas9 strategy was used to edit the chromosome, including deleting or replacing genetic sequences genes in E. coli[55]. Genes idhA, adhE, pflB, poxB, aceE, tktAB, talAB were knocked out. The genes for encoding SBPases from different strains (Bacillus methanolicus, Saccharomyces cerevisiae CEN.PK, Serratia marcescens, Synechocystis PCC 6803, Thermosynechococcus elongatus) and Xfspk from Bifidobacterium longum NCC2705 were synthesized with codon optimization for E. coli (Supplementary Data 7) by BGI Biotechnology and GenScript Biotech Corporation. The SBPase from Synechocystis PCC 6803 was integrated into the genome with a constitutive promoter $P_{j23119}$ and B0034 RBS via CRISPR/Cas9. All the spacers were designed by sgRNAcas9 V3.0[56]. The following concentrations of antibiotics were used: spectinomycin 100 µg/mL, apramycin 100 µg/mL, chloramphenicol 33 µg/mL and tetracycline 17 µg/mL, kanamycin 50 µg/mL and ampicillin 100 µg/mL. To silence genes pfkA, gapA, and gapB, different promoters ($P_{j23116}$, $P_{j23118}$, $P_{j23119}$) and target sequence lengths (0, 50, 80, 100, 150, 200 nt) were selected to verify the effect of asRNA repression. asRNA arrays are assembled through Golden Gate. The mutated glnAp2 promoter containing different enhancers or governors were produced by the one-step site-directed plasmid mutagenesis. Briefly, mutagenesis was generated by inverse PCR of the plasmid using the corresponding primers, and the PCR product digested by DpnI (New England Biolabs, USA) was transformed into E. coli DH5α. Colonies with related mutations were verified by PCR and confirmed by DNA sequencing. To construct a negative feedback circuit, the PhlF repressor induced by promoter Pg was introduced into the genome together with a weak B0031 RBS via CRISPR/Cas9.

### Culture conditions

Luria-Bertani (LB) medium was used for plasmids and strains construction. To test whether SCTPK is functional in vivo, LB medium and modified M9 minimal medium were used, which contains 15.11 g/L $Na_2HPO_4 \cdot 12H_2O$, 3.0 g/L $KH_2PO_4$, 1.0 g/L $NH_4Cl$, 0.5 g/L NaCl, 20 g/L glucose, 1 g/L yeast extract, 0.1 mM $CaCl_2$, 1 mM $MgSO_4$, 0.1 mM $FeCl_3$, and 10 mL of trace element solution (0.18 g/L $ZnSO_4 \cdot 7H_2O$, 0.12 g/L $MnSO_4 \cdot H_2O$, 0.18 g/L $CoCl_2 \cdot 6H_2O$, 0.12 g/L $CuCl_2 \cdot 2H_2O$).

For production of MVA and PHB, the fermentation medium contains 9.8 g/L $K_2HPO_4 \cdot 3H_2O$, 2.1 g/L citrate hydrate, 0.3 g/L ammonium ferric citrate, 0.25 g/L $MgSO_4 \cdot 7H_2O$, 5 g/L beef extract, 2% glucose and 1 mL of trace element solution (0.29 g/L $ZnSO_4 \cdot 7H_2O$, 0.37 g/L $(NH_4)_6Mo_7O_{24} \cdot 4H_2O$, 0.25 g/L $CuSO_4 \cdot 5H_2O$, 1.58 g/L $MnCl_2 \cdot 4H_2O$ and 2.47 g/L $H_3BO_4$). For production of 3HP, the modified minimal medium contains 14 g/L $K_2HPO_4 \cdot 3H_2O$, 5.2 g/L $KH_2PO_4$, 1 g/L NaCl, 1 g/L $NH_4Cl$, 0.25 g/L $MgSO_4 \cdot 7H_2O$, 5 g/L yeast extract and 2% glucose. Shaking flask experiments were carried out in triplicate in 250 mL flasks containing 50 mL medium. To exclude effect of beef extract or yeast extract on production of target chemicals, control experiments were performed using above medium without glucose. The production of 3HP, MVA and PHB was lower than 80 mg/L (Supplementary Fig. 27), indicating that beef extract or yeast extract only had a growth boosting effect, but cannot be used to produce target chemicals. So, yield of target chemical was calculated as the following equation:

$$Yield = produced\ chemical(g)/consumed\ glucose(g) \qquad (1)$$

Fermentation was carried out in a 5-L bioreactor containing 2 L medium. The culture temperature was controlled at 37 °C, the pH was maintained at 7.0, the dissolved oxygen (DO) was kept above 40% saturation via cascade agitation (400–700 rpm), and no chemical inducer was needed. After the initial carbon sources were nearly exhausted, fed-batch mode was commenced by feeding a solution containing 50% (w/v) glucose. Samples were taken as required and analyzed.

## Analytical methods

Cell growth was assayed by measuring optical density (OD) of the culture at 600 nm using a spectrophotometer (U-2900; Hitachi). The residual glucose concentration was detected by an SBA-40ES biological sensing analyzer (Institute of Biology, Shandong Academy of sciences, China). Pyruvate, acetate, lactate, MVA, 3HP were determined using an Agilent 1260 Infinity series HPLC system equipped with an HPX-87H column (Bio-Rad, Hercules, CA) (300 × 7.8 mm) at 40 °C with 5 mmol/L sulfuric acid as the mobile phase. All culture samples were centrifuged at 12,000 × $g$ for 10 min and then filtered through a 0.22 μm filter before analysis.

To determine the AcP content, single colonies harboring the corresponding plasmids were grown overnight in LB medium at 37 °C. Then, the cells were washed, resuspended into fresh M9 medium containing 2% glucose at an initial $OD_{600}$ of 0.02, and grown at 37 °C with shaking. When reaching stationary phase, the cells were washed twice with phosphate buffered saline (PBS, 10 mM, pH7.2–7.4) and disrupted using a high-pressure cell disruptor (Constant Systems LTD, UK). The AcP assay was derived from previous study with minor modifications[12,47]. Specifically, 500 μl of sample was added to 250 μL of hydroxylamine reagent (4 M hydroxylamine hydrochloride: 3.5 M sodium hydroxide = 1:1, v/v), and kept at room temperature for 10 min to form hydroxamate. Then the coloring reagent, 750 μL ferric chloride solution (5% ferric chloride: 12% trichloroacetic acid: 3 M HCl = 1:1:1, v/v/v), was added and kept at room temperature for 5 min, and the absorbance at 540 nm was measured using a microplate reader (Spark, Tecan). PHB was measured using previously reported methods with slight modification[57]. Briefly, cells were separated from the fermentation broth by centrifugation at 12,000 × $g$ and 4 °C for 10 min. Cell pellets were washed twice with distilled water and lyophilized, treated with twenty milliliter of 0.05 M NaOH solution for 3 h and then centrifuged at 15,000 × $g$ for 20 min. The pellet was washed twice with icecold ethanol (95%, analytical grade), and subjected to freeze drying for further analysis.

To detect the release of $CO_2$, a gas analysis system (BCP- $CO_2$, BlueSens, Germany) was used to record and analyze in situ the real-time changes of $CO_2$ in the shake flask during the incubation. Cultures were grown at 37 °C with shaking at 180 rpm. The detector monitors the $CO_2$ volume every 1 min and transmits the information to a computer.

## Protein purification, SDS-PAGE and western blotting

The *E. coli* cells were collected and washed twice with phosphate buffered saline (PBS, 0.01 M, pH 7.2–7.4) and disrupted using a high-pressure cell disruptor (Constant Systems LTD, UK). The $his_6$-tagged recombinant protein was purified using Ni-NTA His·Bind Column (Novagen). The protein was quantified using a Bradford protein assay kit (Beyotime, China), fractionated on a 12% SDS-PAGE gel and visualized by Coomassie blue staining. For western blot, proteins were transferred to PVDF membrane (Millipore) and blocked using quick block western reagent (Beyotime, China). HRP Anti-6X His tag antibody [GT359] (Abcam, catalog No. ab184607, 1:10,000) was used to detect $His_6$-tagged proteins, and acetyl lysine mouse monoclonal antibody (EasyBio, China, catalog No. BE3411, 1:2000) and Goat anti-mouse IgG (H&L) HRP-conjugated antibody (EasyBio, China, catalog No. BE0102, 1:10,000) was used for protein acetylation analysis. Then protein signal was detected using Immobilon Western HRP substrate (Millipore) and Fusion FX6 Imaging System (Vilber, France).

## Enzyme activity assay

The purified protein was desalted using NAP-10 column (GE Healthcare Life Sciences, Pittsburgh, USA). SBP/FBPase activity was determined as following[58]: the reaction mixture containing 50 mM Tris/HCl, pH 8.0, 15 mM $MgCl_2$, 10 mM DTT, 10 mM E4P, 10 mM DHAP and purified FbaA and SBP/FBPase was incubated at 37 °C for 30 min, and the reaction was stopped by addition of 0.2 M perchloric acid. Then the samples were centrifuged and the supernatant was subjected to phosphate quantification. Aliquots (20 μL) of the sample and standards (0-0.5 mM $KH_2PO_4$) were incubated with 340 μL molybdate solution (0.3% ammonium molybdate in 0.55 M $H_2SO_4$) for 10 min at room temperature. Sixty microliters of malachite green solution (0.035% malachite green, 0.35% polyvinyl alcohol) was added and the samples were incubated for further 45 min. The absorbance at 620 nm was determined using a multimode microplate reader (Spark, Tecan). The reaction catalyzed by phosphoketolase took place for 30 min in buffer containing 50 mM Tris/HCl (pH 8.0), 15 mM $MgCl_2$, 10 mM DTT, 5 mM KCl, 20 mM potassium phosphate, 1 mM thiamine pyrophosphate, 0.25 μM phosphoketolase and 10 mM substrate. For the S7P phosphoketolase activity, 10 mM DHAP and E4P were supplemented with 0.25 μM of FbaA and SBPase. For the F6P and X5P phosphoketolase activity, 10 mM F6P and X5P were used, respectively. The reaction was stopped by adding 25 μL of hydroxylamine reagent. Afterwards, AcP concentration was measured.

## Fluorescence assay

To assay cell growth and fluorescence intensity, the corresponding data were recorded on a multimode microplate reader (Spark, Tecan) using 96-well black plates. The mTagBFP2 intensity was measured at an excitation wavelength (EX) of 402 ± 5 nm and an emission wavelength (EM) of 457 ± 10 nm, the GFP intensity was measured at an EX of 480 ± 5 nm and an EM of 520 ± 10 nm, and the mCherry intensity was measured at an EX of 588 ± 10 nm and an EM of 645 ± 10 nm. The background fluorescence of the strain without fluorescent protein expression ($FL_{bg}$) and the background OD of the medium ($OD_{bg}$) were measured, and the relative fluorescence intensities was calculated as the following equation:

$$(FL/OD)_{corrected} = (FL - FL_{bg})/(OD - OD_{bg}) \qquad (2)$$

For the fluorescence microscopy, Laser Scanning Confocal Microscope LSM900 (ZEISS, Germany) was used, and Fiji (NIH) was used to merge the images of the channels. To detect spatial patterning, strain Q4602 was spread onto LB agar plate, and sterilized filter paper disk was placed at the center of the plate and AcP solution was delivered to the disk. After grown overnight at 37 °C, the fluorescence was visualized using a LED transilluminator (BL-20, LABGIC).

## Real-time PCR

Total RNA was isolated using the EASYSpin Plus Bacterial RNA kit (Aidlab Biotechnologies, China), and measured using a Nanodrop one$^C$ spectrophotometer (Thermo Fisher Scientific, Waltham, MA, USA). Genomic DNA was removed and cDNA was obtained by using the Evo M-MLV RT Kit with gDNA Clean for qPRC II (Accurate Biology, China). Quantitative PCR was performed using SYBR Green Pro Taq HS qPCR Kit (Accurate Biology, China) with the QuantStudio 1 system (Applied Biosystems). The constitutively transcribed gene *rpoD* was used as a reference control to normalize total RNA quantity of different samples. The relative amount of mRNA level was calculated using the $2^{-\Delta\Delta Ct}$ method. Three independent biological samples with two technical repeats for each sample were performed.

## Software and statistics

Software for initial data processing was Microsoft Excel 2019, and subsequent analyses and plotting were carried out using OriginPro 2022 (OriginLab) and Graphpad Prism 9 (Graphpad). The spacers of gRNA were designed by sgRNAcas9 v3.0. The fluorescence intensities were determined by using magellan 3.0 (Tecan). The imaging data were obtained and processed by Zen 3.3 (Zeiss) and Fiji (NIH). The relevant curves were fitted using OriginPro 2022 (OriginLab). The logistic model was used to calculate the specific growth rate. Two-tailed Student's *t*-tests were performed to determine the statistical significance.

**Reporting summary**

Further information on research design is available in the Nature Portfolio Reporting Summary linked to this article.

## Data availability

The following information were provided in Supplementary Data files: the plasmids used in this study (Supplementary Data 2), strains (Supplementary Data 3), primers (Supplementary Data 4), the sequences of promoters, governors, enhancers, terminators, linker peptides and genes used in AcP-sensing genetic circuits and biosynthesis (Supplementary Data 5 and 6), sequences and accession numbers of SBPases (Supplementary Data 7). Source data are provided with this paper.

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

## Acknowledgements

This study was financially supported by National Key Research and Development Program of China (2022YFC2104700 and 2021YFC2100500 to G.Z.), NSFC (22277068 to M.L., 32170085 to G.Z.), the Fundamental Research Funds for the Central Universities (2021JCG025 to G.Z.), Distinguished Scholars Program of Shandong University (G.Z.), and Foundation for Innovative Research Groups of State Key Laboratory of Microbial Technology (WZCX2021-02).

## Author contributions

G.Z. and L.G. designed the experiments. L.G., M.L., and Y.B. performed the experiments. G.Z., L.G., M.L., Q.Q., and M.X. analyzed the results. G.Z., L.G., and M.L. wrote the manuscript. All authors edited the manuscript before submission.

## Competing interests

This work has been included in Chinese patent applications by Shandong University (Application No. 2023107236122 and 2023107489681). Co-authors of the patent include G.Z., L.G., M.L., Y.B., Jichao Wang. The patent covers the construction and optimization of the carbon conservation pathway SCTPK, the design and characterization of genetic circuits for activation and inhibition of acetylphosphate, the establishment of a central oscillatory system for acetylphosphate, and the application of the acetylphosphate oscillatory system in the biosynthesis of chemicals and materials. Other authors don't claim competing interests.
