## [Peer Review File · Nature Communications]

Using a synthetic machinery to improve carbon yield with acetylphosphate as the coreReviewers' Comments:

Reviewer #1:

Remarks to the Author:

In this manuscript, titled "A synthetic machinery to improve carbon yield with acetylphosphate as the core," the authors present an artificial pathway, termed SCTPK, controlled by trifunctional phosphoketolase (Xfspk), which effectively reduces carbon loss in the general pathway. Initially, they designed and constructed a SBP-dependent cycle incorporating Xfspk, eliminating the need for carbon rearrangements. Subsequently, the SCTPK pathway was further optimized through combinatorial knockdown using antisense RNA.

Furthermore, the authors developed dynamic genetic circuits regulated by AcP. These circuits facilitated the development of AcP-centered oscillatory systems, enabling the maintenance of intracellular AcP homeostasis and the optimal allocation of cellular resources between cell growth and the synthesis of target chemicals.

The practical applicability of the AcP-based system was successfully demonstrated for the production of various valuable chemicals, such as mevalonate, 3-hydroxypropionic acid, and polyhydroxybutyrate. Remarkably, the yield of mevalonate surpassed its native theoretical limit, and the titer and yield of 3-hydroxypropionic acid exceeded previously reported values achieved using the malonyl pathway.

The manuscript exhibits well-described and logically designed experiments. The results signify significant advancements compared to prior studies, highlighting the potential for broader applications in diverse chemical productions. Nonetheless, several issues require improvement and clarification.

Major comments:

Throughout the manuscript, it is necessary to provide information on the relative strengths of the Anderson promoters used in this study, as this would enhance the manuscript's readability.

Concerning the *in vivo* AcP concentration measurement, it is important to address the issue of variation in AcP levels throughout different cell growth phases. To resolve this concern adequately, a more detailed description of the experimental procedures should be provided, specifying the specific time points at which cells were sampled for AcP measurement. Furthermore, the authors should discuss the potential implications of these sampling time points on the interpretation of the results.

The sampling intervals in certain figures, such as Figs. 5i and 6h, may not adequately capture the oscillatory behavior of the values. For a more convincing demonstration of the oscillatory patterns, it is advisable for the authors to conduct a more comprehensive and tightly spaced sampling of these values.

On page 10, line 193, to enhance the comprehensiveness of the study, it would be advantageous for the authors to provide the overall net reaction of the SCTPK and NOG pathways. Additionally, they should engage in a discussion regarding the similarities and differences between these two pathways.

On page 31, line 632, briefly mentioning the rationale behind utilizing the segmental expression of MCR would provide readers with a better understanding of the manuscript.

On page 32, line 642, it is necessary to explain the reason for changing the base strain selection for 3HP production from a strain with deletions in *pf1B*, *ldhA*, *adhE*, and *poxB* genes to the wild-type BW25113 strain.

On page 35, line 704, please provide the specific value for the native theoretical yield of mevalonate.

On page 38, line 782, the authors should discuss the major sources of ATP and reducing power required for the biosynthesis of 3HP, MVL, and PHB in this study. Moreover, it is crucial to thoroughly investigate potential interactions of native pathways with the introduced SCTPK pathway and their impact on overall 3HP production.

TYPOS: check throughout the manuscript

Page 12, line 232: Please revise "int" to "in".

Page 31, line 622: Please revise "Mevalonate" to "mevalonate".

Page 39, line 790: Please revise "eletrosynthesis" to "electrosynthesis".

Page 40, line 830: Please revise "posttranslantional" to "posttranslational".

Reviewer #2:

Remarks to the Author:

Review:

A synthetic machinery to improve carbon yield with acetylphosphate as the core

Summary:

The manuscript describes the construction of a carbon-conserving SCTPK pathway, which converts glucose into acetyl phosphate (AcP) without carbon loss. The authors also reported an oscillatory system to regulate gene expression in response to the AcP level. The authors showed that integrating the SCTPK pathway with the oscillatory system improves the production and yield of chemicals such as 3-hydroxypropionate (3HP), mevalonate (MVA), and polyhydroxybutyrate (PHB). The results, if verified, are interesting and may lead to potential applications.

However, some experimental results seem to deviate from existing literature results. Also, the calculations of product yield per glucose are incorrect since the production medium contains beef extract or yeast extract, and glucose is not the sole carbon source. The authors should address this concern and perform the calculations using a glucose minimal medium to accurately assess product yield.

This manuscript describes a comprehensive study of three related ideas all linked to Acp: non-oxidative degradation of glucose, dynamic control based on Acp, and oscillatory circuit based on Acp. As these ideas have all been explored, the authors should present the history of the literature to put ideas in proper perspective.

Major comments

1. The origins of ideas of this work are not properly stated. First, the pathway that the authors utilize is a previously proposed pathway (See ref 13 and 14 cited by the authors), but the authors did not give due credit to the existing literature and misguided the readers as if this is a novel pathway invented by the authors. Hellgren et al. (Met Eng, 2020) already demonstrated the pathway (named GATHCYC pathway) in *Saccharomyces cerevisiae*.

2. Second, the glnAP2-AcP dynamic control has been reported in the literature (ref 29), but the authors implied a different story. The concept of AcP gene circuit is not original as it is portrayed in the manuscript. The following statements in the manuscript are incorrect and misleading.

Ln 103: "Although various biosensors were developed recently, there are few reports on the global dynamic regulation of central metabolic networks except those in response to pyruvate¹⁶, fructose-1,6-diphosphate (FBP)¹⁷, E4P and glycerate-3-phosphate¹⁸". This statement fails to acknowledge that AcP control was indeed the first dynamic control reported (ref 29), prior to all those mentioned above by more than 20 years.

Line 313: ",.....that controls a transcriptional response to nitrogen starvation but plays a negligible role under most bioreactor conditions²⁹." This statement failed to state that the same idea for dynamic control for product formation was already reported in ref. 29.

3. Third, the oscillation circuits in this manuscript use the same conceptual design principle as in ref. 52. Both involve oscillation between two metabolites Acp and Acetyl-coA. Both use Acp driven glnAP2

as the positive loop. Both use *glnAP2* to express a repressor to generate a negative control loop, *lacI* in ref 52, *PhIF* in this manuscript. Given such high similarity, reference 52 was cited in line 799 as "Compared with the previously constructed NRI-dependent AcP sensors^{52, 53}, our genetic circuits were optimized thoroughly by adjusting the affinity of activator NRI or repressor *PhIF* to corresponding promoters..." and failed to acknowledge the root of the idea.

4. Line 83-86: The authors' claim that the depletion of glyceraldehyde-3-phosphate (GAP) and the accumulation of sedoheptulose-7-phosphate (S7P) are bottlenecks in the previously demonstrated Non-Oxidative Glycolysis (NOG) pathway is incorrect. The argument is based on simulation results reported in the literature. Failure to simulate the pathway using kinetic models only shows the deficiency of the model, not the design. In fact, an *E. coli* NOG strain has been constructed, and no S7P bottleneck was reported. The authors should properly represent the literature. An alternative form of NOG has value in itself, and should not be built on the basis of an incorrect argument.

5. Although the authors claimed that their pathway is more efficient than the previously published NOG pathway (Ln 88, by citing those two references in which the claim is only based on theoretical calculations), they did not show any proof.

6. Since energy yield is also important for pathway efficiency, it is recommended that the authors include ATP in Fig. 1a to provide a comprehensive assessment of SCTPK.

7. SCTPK contains an ATP futile cycle between F6P and FBP (involving *pfkA* and *glpX*). While the authors have utilized *asRNA* to repress *pfkA* expression, it is important to consider the potential contribution of remaining *pfkA* activity and its isoenzyme, *pfkB*, to this futile cycle. Since futile cycles typically impact cell growth and can lower carbon and energy efficiency, it is necessary for the authors to provide further explanation on how SCTPK achieves higher efficiency despite the presence of this futile cycle.

8. In Fig. 1d, the Q3952 strain (black color) is labeled as Δ tktAB Δ talAB. According to Zhao and Winkler's study published in the *Journal of Bacteriology* in 1994, this strain should require pyridoxine (or 4-hydroxy-L-threonine or glycolaldehyde), aromatic amino acids, and vitamins for growth. Therefore, it is unclear how this strain can grow in the minimal medium. The authors should provide an explanation or clarification for this discrepancy.

9. The authors did not specify their media components of growth rescue experiments ("minimal media" and "rich media", LN200).

10. In Fig. 2c and f, the authors attempted to evaluate the effectiveness of different *asRNA* combinations in enhancing the SCTPK pathway by measuring intracellular acetyl phosphate (AcP) concentrations. However, it should be noted that AcP is an intracellular intermediate that can be readily converted to acetyl-CoA or acetate. As a result, relying solely on AcP concentration may not provide a comprehensive understanding of the pathway's efficiency. It would be valuable if the authors could provide additional explanations or discuss alternative approaches to assess the performance of the SCTPK pathway.

11. The concentrations of acetyl phosphate (AcP) and acetate in the wild-type (WT) strain and Q4531 strain in Fig. 2f and 2k appear to be inconsistent. While Q4531 has a higher AcP concentration, it shows a much lower acetate concentration compared to the WT strain. This observation raises questions about the relationship between AcP and acetate production. It would be helpful if the authors could provide a more detailed explanation or discuss potential factors that could contribute to this discrepancy.

12. Line 300: The authors' conclusion that "combinational *asRNA* arrays have further pulled the carbon flux of competing metabolism into the designed SCTPK..." is based on higher AcP production and lower CO₂ emission as evidence. However, these results do not directly prove the redirection of carbon flux. To provide more solid evidence and support their conclusion, the authors could consider conducting a labeling experiment and calculating the carbon balance. This approach would enable a direct assessment of the carbon flux distribution and confirm the effective redirection of carbon towards the SCTPK pathway. Including such experiments would strengthen the validity of their claims and provide a more comprehensive understanding of the metabolic changes occurring in the engineered strains.

13. In the oscillation circuit, the negative loop was used to control the input of AcP, while the positive loop was to control the output of AcP. This configuration could still reach a steady state at certain AcP level, where the input flux of AcP equals the output flux of AcP. Fig. 5a and Fig. 6a do not explain the

cause of oscillation. They only show what happened if there is oscillation.

14. Regarding oscillatory production, the authors should clearly explain why oscillation can enhance production. Since the circuit only oscillates at the transcription level, proteins may be quite stable. What are the explanations for the enhancement of production?

15. Fig. 6f shows that Acp fluctuation correlates with the glucose concentration. Would it be possible that the "oscillation" is just a result of the cell reacting to the glucose concentration due to the fed batch experiment? More experiments are required to support the authors' claim.

16. The enhancement of production may also come from more optimal expression of key enzymes, and nothing to do with oscillation itself. The period of oscillation is about 10 hours. How does the cell benefit from this oscillation for production?

17. As the AcP concentrations only fluctuate at micromol/OD level (This unit should be converted to molarity), it is hard to justify the enhancement in both growth and production.

18. It is puzzling that the authors chose to use the malonyl-CoA pathway to produce 3-HP and prove their concept of saving carbon through the SCTPK pathway. 3-HP can be produced by a C3 pathway without carbon loss (See Kim et al, 2020, <https://doi.org/10.1002/bit.27344>). Given that the C3 pathway has an even higher titer & yield than the authors achieved, what is the purpose here? Are there any potential benefits of using the malonyl-CoA pathway?

19. From Line 704: The authors should provide the native theoretical yield, and the engineered theoretical yield, and how much percentage their yield exceeded regarding mevalonate production. Similarly, from Line 706: They should also specify the value of "native theoretical yield" of PHB in the text.

20. The production media used in this study are not defined media. They either contain beef extract or yeast extract. Since glucose is not the sole carbon source for 3HP, MVA, and PHB, the carbon yield calculations for each product based on these media are incorrect. To accurately determine the carbon yield, the authors should use a glucose minimal medium for the production experiments. This will ensure that the carbon source is solely derived from glucose and provide a more accurate assessment of the carbon efficiency of the engineered strains.

Minor comments

1. In introduction, the sentence "A strategy to rewrite the central metabolism is the rewiring of the metabolic network" is not clear.

2. Line 768: Should be "PfkA."

3. The authors should provide a graph of fully stoichiometry balanced pathway, including the ATP and NADH balance.

4. Fig 1 d, e, f : The caption sizes are too small. Pls adjust

5. Fig 1e. The gnd KO icon is in the wrong position. It should be the enzyme to convert 6PG to Ru5P.

6. Fig 3d: Please specify what pRSF plasmid is in the figure legend.

7. Fig 5j: The concentration of Acp should be specified in the graph

8. Ln 232 typo: "int"

9. Ln 428: Typo: "An" extra copy

Reviewer #3:

Remarks to the Author:

The authors established a synthetic machinery to enhance carbon yield using acetylphosphate (AcP) as the core. They introduced the carbon-conserving pathway SCTPK into *E. coli*, which can convert glucose into stoichiometric amounts of AcP without carbon loss. Additionally, they developed genetic circuits that positively and negatively respond to AcP and wired them to generate oscillatory systems. Finally, they demonstrated efficient biosynthesis of several chemicals using their synthetic machinery. Overall, the work presented here is interesting and well-conducted, but some issues need to be addressed before publication.

1. To improve the readability of the article, it is suggested to include an overall framework diagram of the study after the introduction section. Additionally, in the schematic diagram of the SCTPK pathway, it would be helpful to label the number of carbon atoms (Cn).
2. In my opinion, "rewiring", "reallocation", and "reconfigured" do not seem to have much difference in engineering central metabolism. However, in the abstract, they are described as two different strategies. Therefore, the storyline in the abstract needs to be reorganized.
3. There are several issues that need to be addressed regarding Fig. 3. (1) in micrograph ii of panel 1b, it is unclear why the color changes after merging, as it appears that mCherry is not expressed. (2) in panel 1d, the original governor and enhancer sequences should also be provided. (3) in panel 1i, the meaning of NI, PI, and other abbreviations should be explained, and the two y-axes should be swapped.
4. Line 410, it is unclear which of the mutated promoters shown in Fig. 3 was used. It appears that the "glnAp2 with mutated governor sequence" (AII) was used here. It is not clear why the authors did not use the glnAp2 with mutated enhancers (AIII), which has a higher dynamic range and lower basal leakage expression.
5. The carbon-conserving pathway consisting of Xfspk and SBPase has already been established in yeast (10.1016/j.ymben.2020.09.003), it is appropriate to cite the previous study when referring to the source of Xfspk (e.g., line97 and line182).

Other general comments

1. To avoid ambiguity, it is recommended to avoid using overly long sentences, such as those in lines 86-91.
2. Supplementary Figs. 21-23 should be reorganized to match the order in which they are presented in the main text.
3. The Discussion section is too lengthy and needs to be condensed, keeping in mind the requirement of a main text of no more than 5000 words. Additionally, the discussion of the results should include relevant Fig. to support the arguments.
4. Spelling errors, such as "dihydroxyacetone," should be identified and rectified in the text.

Response to the reviewers' comments:

We thank the reviewers for their constructive comments and appreciation for the importance of our study: "*The manuscript exhibits well-described and logically designed experiments. The results signify significant advancements compared to prior studies, highlighting the potential for broader applications in diverse chemical productions.*" (Reviewer 1); "*The results, if verified, are interesting and may lead to potential applications.*" (Reviewer 2); "*Overall, the work presented here is interesting and well-conducted.*" (Reviewer 3).

We responded to the Reviewers' comments in a point-by-point form.

We appreciate the opportunity to re-submit our study to Nature Communications and are happy to make any additional modifications suggested by the Editors and Reviewers.

Reviewer #1

1. Throughout the manuscript, it is necessary to provide information on the relative strengths of the Anderson promoters used in this study, as this would enhance the manuscript's readability.

Reply: The relative strengths of the Anderson promoters were determined using GFP as a reporter, and labeled in Fig 3b. The detailed data were shown in Supplementary Fig. 5.

2. Concerning the in vivo AcP concentration measurement, it is important to address the issue of variation in AcP levels throughout different cell growth phases. To resolve this concern adequately, a more detailed description of the experimental procedures should be provided, specifying the specific time points at which cells were sampled for AcP measurement. Furthermore, the authors should discuss the potential implications of these sampling time points on the interpretation of the results.

Reply: As reported previously, *E. coli* intracellular AcP concentration varies with

the growth phases, and the stationary-phase cells growing on glucose present relatively stable level of intracellular AcP, which is 15-20 times higher than that of exponentially growing cells (Mol Cell, 51: 265). So, we determined the AcP concentration using *E. coli* cells in stationary phase, and more detailed experimental procedures were provided in the revised manuscript (page 37, line 773-780).

3. The sampling intervals in certain figures, such as Figs. 5i and 6h, may not adequately capture the oscillatory behavior of the values. For a more convincing demonstration of the oscillatory patterns, it is advisable for the authors to conduct a more comprehensive and tightly spaced sampling of these values.

Reply: Following your suggestion, we have shortened the sampling intervals, determined the intracellular AcP concentration and relative mRNA level of *xfspk* and *pta*, and updated the related figures Fig 6g, 6i, 7g, and 7h. The AcP level in the strain Q4602 carrying SCTPK and Oscillator III fluctuated around 2 mM/OD, which was much lower than that in the strain Q4531 carrying SCTPK alone. The genes *pta* and *xfspk* were transcribed oscillatory in opposite phases, and the mRNA level of *pta* changed synchronously with intracellular AcP concentration. These results were consistent with our previous results, and suggesting that our constructed oscillatory network is capable of regulating expression of metabolic genes precisely, spontaneously, and real-time to maintain the homeostasis of AcP in *E. coli* cells.

4. On page 10, line 193, to enhance the comprehensiveness of the study, it would be advantageous for the authors to provide the overall net reaction of the SCTPK and NOG pathways. Additionally, they should engage in a discussion regarding the similarities and differences between these two pathways.

Reply: Thanks for your suggestion. The similarities and differences between NOG and SCTPK was discussed in the revised manuscript (page 6, line 87-99). Both pathways can convert each mol glucose into 3 mol AcP with consumption of 1 mol ATP, and there is no net production or consumption of reducing power.

On the other hand, SCTPK employs a characteristic phosphoketolase Xfspk acting on three substrates S7P, F6P, and X5P, but the one in NOG (FxpK) catalyzes X5P and F6P. Up to date, SCTPK is the shortest carbon conservation pathway, requiring only 6 enzymes while 8 enzymes in NOG. The SCTPK no longer requires transaldolase and transketolase for carbon rearrangement as NOG pathway, reducing the number of involved enzymes and the complexity of this pathway, representing the carbon atom economy as well as the protein economy. The overall net reactions, related genes, and key enzymes of NOG and SCTPK were listed in Supplementary Table 7.

5. On page 31, line 632, briefly mentioning the rationale behind utilizing the segmental expression of MCR would provide readers with a better understanding of the manuscript.

Reply: MCR is the key enzyme in 3HP biosynthesis via malonyl-CoA pathway, and it catalyzes the two-step reduction of malonyl-CoA with NADPH to 3HP with malonate semialdehyde as intermediate. Our previous study demonstrated that the N- and C-terminal fragments of MCR protein are functionally distinct. MCR-C catalyzes the reduction of malonyl-CoA into malonate semialdehyde, which is further reduced to 3HP by MCR-N. Our results also showed that segmental expression of MCR led to higher overall enzymatic activity and improved 3HP production. (PLoS One, 2013, 8: e75554) So, dissected genes *mcr-n* and *mcr-c* were used in our 3HP-producing strains. This information has been added in the revised manuscript (page 24-25, line 503-508).

*6. On page 32, line 642, it is necessary to explain the reason for changing the base strain selection for 3HP production from a strain with deletions in *pflB*, *ldhA*, *adhE*, and *poxB* genes to the wild-type BW25113 strain.*

Reply: Thanks for your suggestion, and we also realized that the wild-type BW25113 strain was inappropriate as a control. In the revised manuscript, the strain Q3952 (BW25113 *poxB ldhA adhE pflB tktAB talAB*), which was the starting strain for the construction of SCTPK, was also used as the base strain

for the production of 3HP, MVA and PHB, and the results were added in the main text and Fig 7. Furthermore, the productions from BW25113 and Q3952 were compared, showing that deletion of genes involved in by-products and carbon rearrangement did not improve the production of target chemicals significantly.

7. On page 35, line 704, please provide the specific value for the native theoretical yield of mevalonate.

Reply: The native theoretical yield of mevalonate from glucose was added in the revised manuscript (page 27, line 558).

8. On page 38, line 782, the authors should discuss the major sources of ATP and reducing power required for the biosynthesis of 3HP, MVL, and PHB in this study. Moreover, it is crucial to thoroughly investigate potential interactions of native pathways with the introduced SCTPK pathway and their impact on overall 3HP production.

Reply: To increase the carbon yield, we constructed the SCTPK, which converts glucose into stoichiometric amounts of AcP. However, it can not produce ATP and reducing power, those are required for cell growth and production of chemicals such as 3HP, MVA, and PHB.

In our strain, *aceE* gene was knocked out to block acetyl-CoA production from pyruvate decarboxylase, and *tktAB* and *talAB* genes were deleted to block the carbon rearrangement and to exclude the interference of NOG. So, *E. coli* native pathways for glucose catabolism including glycolysis, Entner-Doudoroff pathway, and residual upper section of phosphate pentose pathway are still functional, and can produce energy and reducing equivalents. Furthermore, cells can generate ATP and NAD(P)H via the TCA cycle and respiration. Therefore, we believe that ATP and reducing power required for cell growth and biosynthesis are supplied by *E. coli* native pathways, and main role of SCTPK is to produce AcP from glucose without CO₂ release to improve the production and yield of acetyl-CoA-derived chemicals. This situation is similar to

recombinant strain carrying NOG reported previously (PNAS, 115: 3538).

Currently there are several hybrid routes being established by simultaneous overexpression of NOG and basal central metabolism of *E. coli*. These strategies essentially replenish energy with carbon atom, and this stitching practice results in a theoretical carbon yield of 61.1-83.3%. Now electrical power can be produced in sustainable ways, and can be used to supply reducing equivalent and energy in microbial electrosynthesis system. Application of electrical driven generation of ATP and reducing equivalent in carbon conserving platform strains is believed to be an effective solution. This part has been added in the revised manuscript (page 30-31, line 621-640).

9. TYPOS: check throughout the manuscript. Page 12, line 232: Please revise “int” to “in”. Page 31, line 622: Please revise “Mevalonate” to “mevalonate”. Page 39, line 790: Please revise “eletrosynthesis” to “electrosynthesis”. Page 40, line 830: Please revise “posttranslantional” to “posttranslational”.

Reply: It has been corrected in the revised manuscript:

page 11, line 215, “int” to “in”

page 24, line 496, “Mevalonate” to “MVA” due to the above abbreviation (page 8, line 144)

page 31, line 638, “eletrosynthesis” to “electrosynthesis”

page 40, line 830, in the original manuscript, which has been removed in the revised manuscript.

Reviewer #2

1. The origins of ideas of this work are not properly stated. First, the pathway that the authors utilize is a previously proposed pathway (See ref 13 and 14 cited by the authors), but the authors did not give due credit to the existing literature and misguided the readers as if this is a novel pathway invented by the authors. Hellgren et al. (Met Eng, 2020) already demonstrated the pathway (named GATHCYC pathway) in

Saccharomyces cerevisiae.

Reply: Thanks a lot for your comment. In the revised manuscript, we have rewritten the related part of the Introduction section, and clearly stated that the carbon conserving pathway evaluated here was proposed based on mathematical modeling by Andersen *et al.* (IEEE/ACM Trans Comput Biol Bioinform, 16: 510) and tested in *S. cerevisiae* by Hellgren and colleagues (Metab Eng, 62: 150). However, the previous study was just a proof-of-concept and the engineered yeast strain carrying this pathway still presented low yield of acetyl-CoA derived chemical from glucose. As *E. coli* is also a host microorganism used widely in bioproduction, this SBP-dependent cycle was constructed and optimized thoroughly in *E. coli*. Finally, using our synthetic machinery (consisting of the SBP cycle and AcP-centered oscillator), the highest production and yield of 3HP via malonyl-CoA pathway to date were achieved, and MVA was produced with a yield of 0.61 g/g glucose, exceeding its native theoretical yield (0.54 g/g).

2. *Second, the glnAP2-AcP dynamic control has been reported in the literature (ref 29), but the authors implied a different story. The concept of AcP gene circuit is not original as it is portrayed in the manuscript. The following statements in the manuscript are incorrect and misleading.*

Ln 103: “Although various biosensors were developed recently, there are few reports on the global dynamic regulation of central metabolic networks except those in response to pyruvate¹⁶, fructose-1,6-diphosphate (FBP)¹⁷, E4P and glycerate-3-phosphate¹⁸”. This statement fails to acknowledge that AcP control was indeed the first dynamic control reported (ref 29), prior to all those mentioned above by more than 20 years.

Line 313: “,.....that controls a transcriptional response to nitrogen starvation but plays a negligible role under most bioreactor conditions²⁹.” This statement failed to state that the same idea for dynamic control for product formation was already reported in ref. 29.

Reply: Thanks for your comments. The incorrect and misleading part about the

AcP dynamic control has been rewritten in the revised manuscript (page 7, line 108-125).

In the Introduction section, we introduced the development history of the *glnAp2*-AcP dynamic control. It was reported that AcP can donate its phosphoryl group to and activate response regulators of two-component signal transduction systems. The Ntr system, consisting of the sensor NRII (*glnL* product) and the response regulator NRI (*glnG* product), controls a transcriptional response to nitrogen starvation and plays a negligible role under most bioreactor conditions. As NRI itself can sense the AcP level and activate the *glnAp2* promoter when NRII is absent, Farmer *et al.* knocked out NRII and recruited NRI as a dynamic controller to activate the expression of lycopene biosynthetic genes from *glnAp2* promoter in response to the glycolytic flux indicator AcP in engineered *E. coli* strain, resulting in improved lycopene production (Nat Biotechnol, 18: 533, ref 29 in the original manuscript). This was the first example of dynamic control based on sensing of a certain metabolite.

3. Third, the oscillation circuits in this manuscript use the same conceptual design principle as in ref. 52. Both involve oscillation between two metabolites Acp and Acetyl-coA. Both use Acp driven glnAP2 as the positive loop. Both use glnAP2 to express a repressor to generate a negative control loop, lacI in ref 52, PhlF in this manuscript. Given such high similarity, reference 52 was cited in line 799 as “Compared with the previously constructed NRI-dependent AcP sensors^{52, 53}, our genetic circuits were optimized thoroughly by adjusting the affinity of activator NRI or repressor PhlF to corresponding promoters...” and failed to acknowledge the root of the idea.

Reply: In the revised manuscript, we stated clearly that AcP-centered oscillators were designed with the introduction of AcP-induced LacI repressor that would block the expression of NRI from the *glnAp2* promoter (Cell, 113: 597, ref 53 in the original manuscript) or Pta that converts acetyl-CoA into AcP (Nature, 435: 118, ref 52 in the original manuscript). Therefore, we believe that NRI-based genetic circuits with appropriate properties are capable to control expression of related genes in response to AcP level, and will help to maintain the intracellular AcP homeostasis. (page 7, line 125-130)

4. *Line 83-86: The authors' claim that the depletion of glyceraldehyde-3-phosphate (GAP) and the accumulation of sedoheptulose-7-phosphate (S7P) are bottlenecks in the previously demonstrated Non-Oxidative Glycolysis (NOG) pathway is incorrect. The argument is based on simulation results reported in the literature. Failure to simulate the pathway using kinetic models only shows the deficiency of the model, not the design. In fact, an E. coli NOG strain has been constructed, and no S7P bottleneck was reported. The authors should properly represent the literature. An alternative form of NOG has value in itself, and should not be built on the basis of an incorrect argument.*

Reply: It was made clear that Andersen *et al.* proposed a new variant of the NOG pathway with introduction of a novel phosphoketolase activity towards S7P (IEEE/ACM Trans Comput Biol Bioinform, 16: 510). Subsequently, Hellgren *et al.* constructed the proposed NOG variant in *Saccharomyces cerevisiae* and pointed out that there might be some kinetic bottlenecks in NOG pathway based on mathematical modelling (Metab Eng, 62: 150). In the revised manuscript, this sentence has been removed, and we objectively described the results published in previous literatures (page 5, line 83-87).

5. *Although the authors claimed that their pathway is more efficient than the previously published NOG pathway (Ln 88, by citing those two references in which the claim is only based on theoretical calculations), they did not show any proof.*

Reply: In the revised manuscript, this sentence has been removed, and the similarities and differences between NOG and SCTPK was discussed (page 6, line 91-99). Both pathways can convert each mol glucose into 3 mol AcP with consumption of 1 mol ATP, and there is no net production or consumption of reducing power. On the other hand, SCTPK is the shortest carbon conservation pathway, requiring only 6 enzymes while 8 enzymes in NOG. The SCTPK no longer requires transaldolase and transketolase for carbon rearrangement as NOG pathway, reducing the number of involved enzymes and the complexity of this pathway, representing the carbon atom economy as well as the protein economy. The overall net reactions, related genes, and key enzymes of NOG and SCTPK were listed in Supplementary Table 7.

6. *Since energy yield is also important for pathway efficiency, it is recommended that the authors include ATP in Fig. 1a to provide a comprehensive assessment of SCTPK.*

Reply: SCTPK does not produce ATP and reducing power, and one molecule of ATP is consumed in the conversion of glucose into glucose-6-phosphate. This has been added in Fig 2a in the revised manuscript.

7. *SCTPK contains an ATP futile cycle between F6P and FBP (involving *pfkA* and *glpX*). While the authors have utilized *asRNA* to repress *pfkA* expression, it is important to consider the potential contribution of remaining *pfkA* activity and its isoenzyme, *pfkB*, to this futile cycle. Since futile cycles typically impact cell growth and can lower carbon and energy efficiency, it is necessary for the authors to provide further explanation on how SCTPK achieves higher efficiency despite the presence of this futile cycle.*

Reply: As your concern, there is indeed a potential futile cycle between the SCTPK enzyme sedoheptulose/fructose biphosphatase (SyGlpX) and the glycolytic enzyme phosphofructokinase (Pfk). *E. coli* contains two Pfk isozymes PfkA and PfkB. The *pfkB* gene utilizes rare codons and has a poor consensus promoter sequence, suggesting it is a weakly expressed gene (Gene, 28: 337), and only less than 5% of Pfk activity in *E. coli* cells can be attributed to PfkB (Biochim Biophys Acta, 381: 257). So, only the *pfkA* gene was knocked out to remove this potential futile cycle in recombinant *E. coli* strain carrying NOG pathway in previous study (PNAS, 115: 3538). However, it was reported that *pfkA* knockout severely suppressed the cell growth and glucose uptake in various studies (Metab Eng, 68: 142; Eur J Biochem, 270: 880; Metab Eng, 55: 249). Then, expression of PfkA was repressed using CRISPRi system in engineered *E. coli* strain harboring the EP-bifido pathway, leading to a significant improvement of MVA yield (Microb Cell Fact, 20: 32).

Based on these facts, we just knocked down the expression of *pfkA* using antisense RNA (*asRNA*) interference to suppress this potential futile cycle. When *asRNA* targeting *pfkA* was overexpressed, the transcription of *pfkA* was reduced to 27.2% of that of the control strain (Fig. 3e), and the intracellular AcP

level was 2.23-time and 1.24-time higher than the wild-type strain and Q3964, respectively (Fig. 3c). In summary, we believed that the potential futile cycle between SyGlpX and Pfk has been eliminated basically.

8. *In Fig. 1d, the Q3952 strain (black color) is labeled as $\Delta tktAB\Delta talAB$. According to Zhao and Winkler's study published in the Journal of Bacteriology in 1994, this strain should require pyridoxine (or 4-hydroxy-L-threonine or glycolaldehyde), aromatic amino acids, and vitamins for growth. Therefore, it is unclear how this strain can grow in the minimal medium. The authors should provide an explanation or clarification for this discrepancy.*

Reply: As you mentioned, growth of $\Delta tktAB\Delta talAB$ mutant requires pyridoxine, aromatic amino acids and vitamins. In this experiment, M9 minimal medium was supplemented with 1 g/L yeast extract, a complex mixture of amino acids, carbohydrates, vitamins, trace elements, etc. As shown in Fig 2d, supplementation of yeast extract supported the growth of Q3952. The strain Q3952 $\Delta aceE$ can not grow under the same conditions, suggesting that deletion of *aceE* led to depletion of acetyl-CoA. In contrast, the strain Q3964 generated by introduction of Xfspk and SyGlpX into Q3952 $\Delta aceE$ recovered from the growth incapability, demonstrating that the acetyl-CoA pool was replenished by the constructed SCTPK. For a clearer description, we have added detailed medium composition in the revised manuscript (page 10, line 183; page 35, line 742-746).

9. *The authors did not specify their media components of growth rescue experiments (“minimal media” and “rich media”, LN200).*

Reply: As described above, modified M9 medium supplemented with 1 g/L yeast extract and LB medium were used in the growth rescue experiments. We have added detailed medium composition in the revised manuscript (page 10, line 183; page 35, line 742-746).

10. *In Fig. 2c and f, the authors attempted to evaluate the effectiveness of different*

asRNA combinations in enhancing the SCTPK pathway by measuring intracellular acetyl phosphate (AcP) concentrations. However, it should be noted that AcP is an intracellular intermediate that can be readily converted to acetyl-CoA or acetate. As a result, relying solely on AcP concentration may not provide a comprehensive understanding of the pathway's efficiency. It would be valuable if the authors could provide additional explanations or discuss alternative approaches to assess the performance of the SCTPK pathway.

Reply: Thanks for your comment. Since SCTPK can convert glucose into stoichiometric amounts of AcP, whereas the native pathway cannot. Therefore, AcP is one of the most direct indicators to detect the effectiveness of the pathway, which has been used as a major reference in studies constructing different NOG variant pathways (Metab Eng, 62: 150; Metab Eng, 51: 79), and we also used the AcP level to evaluate the effectiveness of different asRNAs. In addition, the cell growth was also used as a reference. If an asRNA expressed from a certain promoter increases the AcP level and does not affect cell growth significantly, we will consider that this asRNA is effective.

To evaluate the final strain Q4531, more analyses were carried out. SCTPK pathway does not produce CO₂, and the total CO₂ release of Q4531 strain decreased to 47.4% compared with wild-type *E. coli* (Fig 3i). Moreover, the strain Q4531 showed a significant decrease in pyruvate concentration relative to the wild-type strain and Q3964 (Fig 3j), suggesting that the metabolic flow more shifted to the SCTPK following gene silencing with combinational asRNA arrays. Furthermore, the strain Q4531 showed an acetate accumulation capacity only representing 11% of the wild-type strain, indicating that the accumulated AcP could not be converted into acetate under this condition. In addition, the high acetylation level of Q4531 proteins (Fig 6h) also indicated the high intracellular AcP concentration. Altogether, combinational asRNA arrays have further pulled the carbon flux of competing metabolism into the designed SCTPK, enhanced the production of AcP and repressed the release of CO₂ and accumulation of byproducts.

11. *The concentrations of acetyl phosphate (AcP) and acetate in the wild-type (WT)*

strain and Q4531 strain in Fig. 2f and 2k appear to be inconsistent. While Q4531 has a higher AcP concentration, it shows a much lower acetate concentration compared to the WT strain. This observation raises questions about the relationship between AcP and acetate production. It would be helpful if the authors could provide a more detailed explanation or discuss potential factors that could contribute to this discrepancy.

Reply: It was reported that an engineered yeast strain carrying hybrid routs of glycolysis and partial NOG presented higher AcP level and higher acetate accumulation (Nature, 537: 694). It was believed that the acetate was produced in two pathways. First, when growing on glucose, both *E. coli* and *Saccharomyces cerevisiae* can uptake and metabolize glucose rapidly, and accumulate a large amount of acetate in the culture broth, which is well-documented and was defined as overflow metabolism or Crabtree effect in yeast. Secondly, acetate arises from hydrolysis of AcP by glycerol-3-phosphate phosphatase Rhr2 and Hor2, and deletion of Rhr2 decreased the acetate production more than 2 folds (Nature, 537: 694).

In our case, Q4531 strain carrying SCTPK showed much higher intracellular AcP concentration than wild-type strain, and generated almost no acetate. These performances were similar to previously reported *E. coli* strains harboring EP-bifido pathway (Metab Eng, 51: 79; Front Bioeng Biotechnology, 8: 517336). This may be caused by the following reasons: First, expression of glycolytic enzymes PfkA and GapA was suppressed using asRNA interference, that will decrease the metabolic flux through glycolysis pathway and then block the overflow metabolism. Secondly, *E. coli* glycerol-3-phosphate phosphatase HxpA does not show AcP hydrolytic activity, and no *E. coli* enzyme with high AcP hydrolytic activity was reported. Thirdly, pyruvate oxidase PoxB catalyzes the oxidative decarboxylation of pyruvate to acetate, and the *poxB* gene was deleted in our strain. Although AckA can also produce acetate from AcP, the AckA enzyme has been observed to be more active in the early exponential phase, and the AckA activity can be inhibited by acidic pH caused by acetate accumulation. In contrast, the PoxB pathway is more active in late exponential and stationary phase, and the PoxB activity is upregulated under acidic conditions (Biotechnol Prog, 21: 1062). Therefore, PoxB is considered as the main pathway for acetate accumulation, and knockout of *poxB* should greatly

contribute to the acetate decrease.

12. Line 300: The authors' conclusion that "combinational asRNA arrays have further pulled the carbon flux of competing metabolism into the designed SCTPK..." is based on higher AcP production and lower CO₂ emission as evidence. However, these results do not directly prove the redirection of carbon flux. To provide more solid evidence and support their conclusion, the authors could consider conducting a labeling experiment and calculating the carbon balance. This approach would enable a direct assessment of the carbon flux distribution and confirm the effective redirection of carbon towards the SCTPK pathway. Including such experiments would strengthen the validity of their claims and provide a more comprehensive understanding of the metabolic changes occurring in the engineered strains.

Reply: Thanks a lot for your comment. In our manuscript, the SCTPK pathway was constructed and optimized in *E. coli* strain. The function of SCTPK was proved in vivo by providing essential metabolic intermediate (acetyl-CoA and ribose-5-phosphate) and recovering auxotrophy strains from growth incapability. In the strain with SCTPK, it was noticed that some enzymes may shunt this pathway and cause carbon loss, including PfkA, GapA, and GapB. Then expression of those three enzymes was suppressed using antisense RNA interference. Finally, the strain Q4531 with highly repressed *pfkA*, moderately repressed *gapA* and weakly repressed *gapB* was the best among all constructed strains, presenting significantly improved AcP production, much lower CO₂ release and accumulation of the byproduct acetate. Hence, it was believed that the optimization has improved the function of SCTPK remarkably. As you suggested, labeling experiment and metabolic flux analysis will indeed be a direct evidence for the carbon flux redistribution from competing pathways into SCTPK, however it is hard for us to finish this experiment in a revision of four weeks. So, we have rewritten this sentence into "combinational asRNA arrays have further enhanced the effectiveness of the designed SCTPK..." in the revised manuscript (page 14, line 262-263).

13. In the oscillation circuit, the negative loop was used to control the input of AcP,

while the positive loop was to control the output of Acp. This configuration could still reach a steady state at certain Acp level, where the input flux of Acp equals the output flux of Acp. Fig. 5a and Fig. 6a do not explain the cause of oscillation. They only show what happened if there is oscillation.

Reply: Theoretically, a steady state will be reached when the input and output of AcP are equal. However, this steady state was not observed experimentally, not only in our study, but also the previously constructed AcP oscillators (Cell, 113: 597; Nature, 435: 118) and malonyl-CoA oscillators (PNAS, 111: 11299; ACS Synth Biol, 9: 1059).

In our design, the oscillator contains an “activator module” consisting of a modified *glnAp2* promoter, driving the expression of *pta* in response to phosphorylated NRI. In the “repressor module”, NRI-induced PhIF repressor will block the transcription of *xfspk* from *phIF* promoter. In other word, AcP concentration determines the mRNA level of related genes of enzyme or regulator, and then proteins translated from those mRNA fulfill their enzymatic or regulatory activities. So, the changes at enzyme activity level lag behind the sensing of AcP concentration, and AcP concentration will vary during the time lag, resulting in the fact that a perfect synchronization of AcP sensing and enzyme activity regulation could hardly be realized. Hence, we think that the theoretically steady state is hardly observed in experiments. For a clearer description, we have revised Fig. 6a.

14. Regarding oscillatory production, the authors should clearly explain why oscillation can enhance production. Since the circuit only oscillates at the transcription level, proteins may be quite stable. What are the explanations for the enhancement of production?

Reply: Currently, the circuits of oscillators at the transcription level include not only the genetic metabolic oscillators in this study, but also synthetic genetic oscillator (Nature, 43: 335), dual-feedback oscillator (Nature 456: 516), etc. Similar with our results shown in Fig 6b-f, these oscillators illustrate the oscillatory behavior by the periodic changes of the fluorescent proteins. This suggests that the protein level can be adjusted by being transcriptionally

regulated by the oscillators. Furthermore, in a recent study (Science, 378: eabk2066), it was shown that protein levels in *E. coli* are predominantly set transcriptionally, with relatively invariant posttranscriptional characteristics (translation efficiencies and degradation rates) for most mRNAs. This indicates that oscillatory regulation dependent on engineered transcription can effectively control the rhythm of changes in protein levels.

In addition to effectively regulating the expression of relevant genes, the genetic metabolic oscillators have been further applied in metabolic engineering and synthetic biology research. Through the design of genetic circuits, the expression of product synthesis genes is automatically turn off when there is insufficient key metabolite and automatically turn on after accumulation. This periodic oscillation allows a rational allocation of cellular resources between cell growth and the synthesis of target chemicals. It avoids the problems of metabolic imbalance and waste of carbon source caused by the overexpression of product synthesis genes despite the insufficient precursor and the scarcity of cellular resources. Also, the oscillator exhibits a bifunctional gene expression pattern that dynamically regulates the source and sink of AcP, which leads to its relative intracellular homeostasis, preventing its excessive accumulation or consumption that could affect the global cellular metabolism. In addition, enhancement of production caused by metabolic oscillators were also reported by different groups (PNAS, 111: 11299; ACS Synth Biol, 9: 1059). Therefore, we believe that the biosynthesis of chemicals and materials can be facilitated by this dynamic pattern of carbon flux reallocation.

15. Fig. 6f shows that Acp fluctuation correlates with the glucose concentration. Would it be possible that the “oscillation” is just a result of the cell reacting to the glucose concentration due to the fed batch experiment? More experiments are required to support the authors’ claim.

Reply: In shaking flask cultivation of strain Q4602 carrying SCTPK and Oscillator III, the glucose concentration should decrease over time due to consumption of the cells, while the AcP level presented a periodic fluctuation. In Fig 7f, fed-batch mode was commenced by feeding a solution containing 50%

(w/v) glucose, causing the varied glucose concentration. As shown in the following figure, there is no significant correlation between glucose concentration and AcP level. So, we don't think that oscillation of AcP level correlates with the glucose concentration.

Fig. The residual glucose concentration and intracellular AcP concentration during 3HP fed-batch fermentation of strain Q4617.

16. *The enhancement of production may also come from more optimal expression of key enzymes, and nothing to do with oscillation itself. The period of oscillation is about 10 hours. How does the cell benefit from this oscillation for production?*

Reply: As you mentioned, optimal expression of key enzymes is essential for high production and yield of target products. In our study, AcP-centered oscillators are designed to regulate expression of key enzymes in AcP production and consumption in response to intracellular AcP level precisely, spontaneously, and real-time. Under control of the oscillator, expression of product synthetic genes can be automatically turned off when there is insufficient key intermediate, and automatically turned on when the key metabolite accumulates. So, metabolic oscillator may offer a way to balance fluxes and minimize protein expression burden when heterologous pathways were expressed in engineered strains. Furthermore, oscillatory gene expression profiles allow trade-offs between growth and production to be better managed and can help avoid build-up of undesired intermediates.

When conducting fermentations of 3HP, MVA, and PHB, in addition to the

oscillation mode of regulation, we also performed control experiments with dynamic activation (switch) regulation. In these two modes of regulation, we kept identically in the expression of the key enzymes.

The following illustrates the regulation of MVA as an example (same situation for 3HP and PHB). Switch I and oscillator I or switch II and oscillator II, which have identical control of the key enzymes, including promoter, RBS, and copy number. The only difference between the two modes is that the oscillation mode simultaneously introduces a negative feedback response system for AcP. This genetic circuit, which responds positively or negatively to AcP, constitutes the gene-metabolic oscillator. The fermentation results showed that the yield and titer of MVA were higher in the oscillation than in the switch. Further maintenance of the oscillatory behavior resulted in a higher yield and titer of MVA, suggesting that production facilitation is associated with oscillation. Similar fermentation results for 3HP and PHB demonstrate the versatility of this oscillation mode.

Oscillatory regulation of key enzymes has been employed in biosynthesis of different products. For example, a kinetic model incorporating oscillatory expression of sets of glycolytic proteins increased phosphoenolpyruvate pools by 1.86-fold (PLoS Comput Biol, 10: e1003658). A similar strategy was recently used by Xu *et al.* to balance malonyl-CoA pools for fatty acid production (PNAS, 111: 11299). Promoters responding to malonyl-CoA were designed based on FapR, a malonyl-CoA responsive transcription factor from *Bacillus subtilis*, which allows both upregulation and downregulation of gene expression when the intracellular malonyl-CoA level increase. Upon accumulation of malonyl-CoA, these promoters decreased expression of the upstream malonyl-CoA production genes and enhanced expression of the downstream consumption operon, resulting in oscillatory levels of intracellular malonyl-CoA with a period of 7 h and a 2.1-fold improvement in fatty acid titers over the unregulated pathway.

17. As the AcP concentrations only fluctuate at micromol/OD level (This unit should be converted to molarity), it is hard to justify the enhancement in both growth and

production.

Reply: Thanks for your suggestion. We have changed the unit of AcP concentration to mM/OD.

18. It is puzzling that the authors chose to use the malonyl-CoA pathway to produce 3-HP and prove their concept of saving carbon through the SCTPK pathway. 3-HP can be produced by a C3 pathway without carbon loss (See Kim et al, 2020, <https://doi.org/10.1002/bit.27344>). Given that the C3 pathway has an even higher titer & yield than the authors achieved, what is the purpose here? Are there any potential benefits of using the malonyl-CoA pathway?

Reply: As mentioned above, 3HP production via a C3 pathway has achieved the highest 3HP titer and yield so far. However, there are some disadvantages in the C3 pathway. First, it is a glycerol-dependent 3HP biosynthetic pathway, and only glycerol can be used as carbon source. Second, it requires an exogenous supply of expensive coenzyme vitamin B12, a cofactor of glycerol dehydratase, increasing the production cost of 3HP from glycerol. Furthermore, the reducing power imbalance was also a burden for cellular metabolism.

In the malonyl-CoA pathway, acetyl-CoA is converted into malonyl-CoA by the acetyl-CoA carboxylase, and malonyl-CoA is reduced to 3HP by malonyl-CoA reductase. In this route, 3HP is derived from core metabolic intermediate, so that various sugars from lignocellulosic biomass can be used as a raw material for 3HP production. Moreover, the conversion of acetyl-CoA into malonyl-CoA is a CO₂ fixation reaction, resulting in higher carbon atom economy of the malonyl-CoA pathway. Additionally, the production of 3HP from glucose is redox neutral. Because of these advantages, the malonyl-CoA pathway was considered as a promising route for 3-HP biosynthesis, and has been successfully constructed in various species including *E. coli*, yeast and cyanobacteria in the recent years (Metab Eng, 34:104; Metab Eng, 22: 104; Metab Eng, 31: 163; Microb Cell Fact, 15: 53; mBio, 5: e01130; Bioresour Technol, 200: 897).

19. *From Line 704: The authors should provide the native theoretical yield, and the engineered theoretical yield, and how much percentage their yield exceeded regarding mevalonate production. Similarly, from Line 706: They should also specify the value of "native theoretical yield" of PHB in the text.*

Reply: As suggested, the native and engineered theoretical yield has been added in the revised manuscript (page 27, line 556-562).

20. *The production media used in this study are not defined media. They either contain beef extract or yeast extract. Since glucose is not the sole carbon source for 3HP, MVA, and PHB, the carbon yield calculations for each product based on these media are incorrect. To accurately determine the carbon yield, the authors should use a glucose minimal medium for the production experiments. This will ensure that the carbon source is solely derived from glucose and provide a more accurate assessment of the carbon efficiency of the engineered strains.*

Reply: Thanks for your comment. In our production of various chemicals and materials such as 3HP, MVA and PHB using the engineered strains, we conducted relevant control experiments to evaluate the contribution of beef extract or yeast extract to the production of each strain. In this experiment, 2% glucose was removed from the fermentation medium, while 0.5% beef extract or yeast extract and all other ingredients were remained. The results showed that the production of 3HP, MVA and PHB at the end of fermentation was lower than 80 mg/L for each strain, and final cell density was higher than OD₆₀₀ of 2. This indicates that 0.5% beef extract or yeast extract only had a growth boosting effect, but cannot be used to produce target chemicals by these recombinant strains. This is a common method usually used in fermentation, which involves supplementation of a small amount of yeast extract or beef extract to promote cell growth, with glucose as the solo carbon source (Green Chem, 23: 8694; Metab Eng, 27: 76; J Biosci Bioeng, 127: 301).

Fig Final cell density and production of 3HP (a), MVA (b) and PHB (c) when grown in fermentation medium without glucose.

21. In introduction, the sentence “A strategy to rewrite the central metabolism is the rewiring of the metabolic network” is not clear.

Reply: The Introduction section has been rewritten in the revised manuscript.

22. Line 768: Should be “PfkA.”

Reply: It has been corrected in the revised manuscript (page 29, line 606).

23. The authors should provide a graph of fully stoichiometry balanced pathway, including the ATP and NADH balance.

Reply: SCTPK does not produce reducing power and consumes one ATP molecule. This has been added in Fig. 2a.

24. Fig 1 d, e, f: The caption sizes are too small. Pls adjust

Reply: Fig. 1 (Fig. 2 in the revised manuscript) has been adjusted as suggested.

25. Fig 1e. The gnd KO icon is in the wrong position. It should be the enzyme to convert 6PG to Ru5P.

Reply: This has been corrected in the revised manuscript.

26. *Fig 3d: Please specify what pRSF plasmid is in the figure legend.*

Reply: pRSF plasmid is a derivative of pRSFDuet-1, in which the *lacI* gene and T7 promoter have been deleted. This information has been added in the legend of Fig 4.

27. *Fig 5j: The concentration of Acp should be specified in the graph*

Reply: In this experiment, a filter paper disk soaked with 15 μ L of 1 M AcP was placed in the center of agar plate to create a radial AcP gradient. This information has been added in the legend of Fig 6.

28. *Ln 232 typo: "int"*

Reply: It has been corrected in the revised manuscript (page 11, line 215).

29. *Ln 428: Typo: "An" extra copy*

Reply: It has been corrected in the revised manuscript (page 18, line 356).

Reviewer #3

1. *To improve the readability of the article, it is suggested to include an overall framework diagram of the study after the introduction section. Additionally, in the schematic diagram of the SCTPK pathway, it would be helpful to label the number of carbon atoms (Cn).*

Reply: Thanks for your suggestion. A diagram displaying the whole concept of this study was added in the revised manuscript as Fig 1, and the numbers of carbon atoms in the intermediates of SCTPK were labeled.

2. *In my opinion, "rewiring", "reallocation", and "reconfigured" do not seem to have*

much difference in engineering central metabolism. However, in the abstract, they are described as two different strategies. Therefore, the storyline in the abstract needs to be reorganized.

Reply: The Abstract section has been rewritten in the revised manuscript.

3. There are several issues that need to be addressed regarding Fig. 3. (1) in micrograph ii of panel 1b, it is unclear why the color changes after merging, as it appears that mCherry is not expressed. (2) in panel 1d, the original governor and enhancer sequences should also be provided. (3) in panel 1i, the meaning of NI, PI, and other abbreviations should be explained, and the two y-axes should be swapped.

Reply: Thank you for your precious comments, and we have addressed the following issues according to your suggestion:

In Fig. 4b, the color change was caused by the image processing software we used before. We performed a batch merge using another program with a different algorithm, and this issue has been corrected.

In Fig.4d, the original sequences of governor and enhancer of *glnAp2* promoter have been added.

For Fig. 4i, the meanings of NI, PI and other abbreviations are explained in the legend, as well as the two y-axes are swapped as suggested.

4. Line 410, it is unclear which of the mutated promoters shown in Fig. 3 was used. It appears that the "glnAp2 with mutated governor sequence" (AII) was used here. It is not clear why the authors did not use the glnAp2 with mutated enhancers (AIII_f), which has a higher dynamic range and lower basal leakage expression.

Reply: Thanks for your comment. When we constructed the repression circuit, we introduced the PhI_f repressor and used *glnAp2* with only governor sequence mutation for the activation of this repressor, allowing the activated repressor to further inhibit the expression of downstream genes. After enhancer is then mutated (IP, used in AIII_f), the promoter, although possessing a higher dynamic range and lower basal leakage expression, has essentially a higher response

threshold than the promoter with only governor mutation at the same *glnG* expression level due to the altered affinity. To accelerate the cascade response of the repression module (Phlf activation followed by repression of downstream gene expression), we selected the promoter with a lower threshold to control Phlf activation, allowing transcription of downstream genes to be rapidly repressed by Phlf. In addition, we considered that in the oscillator, acetyl phosphate may oscillate in a certain range. A relatively lower threshold may allow the gene to remain responsive at lower acetyl phosphate states. Finally, besides the promoter used for Phlf expression, as done in this experiment, the interaction between the repressor and the downstream promoter P_{phlf} can also be further tuned, allowing for further optimization of the genetic circuit.

5. The carbon-conserving pathway consisting of Xfspk and SBPase has already been established in yeast (10.1016/j.ymben.2020.09.003), it is appropriate to cite the previous study when referring to the source of Xfspk (e.g., line97 and line182).

Reply: In the revised manuscript, we have rewritten the related part of the Introduction section, and stated that the carbon conserving pathway evaluated here was proposed based on mathematical modeling by Andersen *et al.* (IEEE/ACM Trans Comput Biol Bioinform, 16: 510) and tested in *S. cerevisiae* by Hellgren and colleagues (Metab Eng, 62: 150).

6. To avoid ambiguity, it is recommended to avoid using overly long sentences, such as those in lines 86-91.

Reply: In the revised manuscript, the sentence in lines 86-91 has been removed, and we have reviewed the whole manuscript to avoid using overly long sentences.

7. Supplementary Figs. 21-23 should be reorganized to match the order in which they are presented in the main text.

Reply: Thanks for your suggestion. We have reordered the Supplementary Figs.

21-23 to match their order in the text. Please see line 540, 547 and 548.

8. The Discussion section is too lengthy and needs to be condensed, keeping in mind the requirement of a main text of no more than 5000 words. Additionally, the discussion of the results should include relevant Fig. to support the arguments.

Reply: Thanks for your comment. We have streamlined some contents of the Discussion section, and cited relevant figures to support the arguments.

9. Spelling errors, such as "dihydroxyacetone," should be identified and rectified in the text.

Reply: It has been corrected in the revised manuscript (page 9, line 159).

Reviewers' Comments:

Reviewer #1:

Remarks to the Author:

Authors revised their manuscript according to the reviewers' comments. I do not have any further comment.

Reviewer #2:

Remarks to the Author:

The revised version was much clearer. I have just one more comment. Since the 3 HP experiment was done in a rich medium, please put in the "no glucose" control as a comparison. Also explain how the yield was calculated in the rich medium, since there are multiple ways to define it.

Reviewer #3:

Remarks to the Author:

The authors have addressed my comments and I recommend acceptance for publication.

Reviewer #2

The revised version was much clearer. I have just one more comment. Since the 3HP experiment was done in a rich medium, please put in the “no glucose” control as a comparison. Also explain how the yield was calculated in the rich medium, since there are multiple ways to define it.

Reply: The results using media without glucose was provided in the revised manuscript as Supplementary Figure 27. The results showed that the production of 3HP, MVA and PHB at the end of fermentation was lower than 80 mg/L for each strain, and final cell density was higher than OD₆₀₀ of 2. This indicates that beef extract or yeast extract only had a growth boosting effect, but cannot be used to produce target chemicals by these recombinant strains. So, the yield of target chemical was calculated as following: yield = produced chemical (g) / consumed glucose (g). This information was also provided in the revised manuscript (page 35, line 749-755).